# Sharpness-Aware Pretraining Mitigates Catastrophic Forgetting

Ishaan Watts [* 1]   Catherine Li [* 1]   Sachin Goyal [1]   Jacob Mitchell Springer [† 1]   Aditi Raghunathan [† 1]

## Abstract

Pretraining optimizers are tuned to produce the strongest possible base model, on the assumption that a stronger starting point yields a stronger model after subsequent changes like post-training and quantization. This overlooks the geometry of the base model which controls how much of the base model's capabilities survive subsequent parameter updates. We study three pretraining optimization approaches that bias optimization toward flatter minima: Sharpness-Aware Minimization (SAM), large learning rates, and shortened learning rate annealing periods. Across model sizes ranging from 20M to 150M parameters, we find that these interventions consistently improve downstream performance after post-training on five common datasets with up to 80% less forgetting. These principles hold at scale: a short SAM mid-training phase applied to an existing OLMo-2-1B checkpoint reduces forgetting by 31% after MetaMath post-training and by 40% after 4-bit quantization.

*Figure 1.* **Main results from OLMo-2-1B experiments.** We take an OLMo-2-1B model pretrained on 4T tokens and then mid-train it for 50B tokens using SAM and AdamW. After further modification by SFT (MetaMath, StackMathQA, Tülu-3, and MusicPile) and 4-bit quantization, SAM reduces forgetting on the pretraining eval benchmark.

## 1. Introduction

Pretraining optimization choices are typically selected to improve base-model quality, as measured by pretraining loss or benchmark performance (OLMo et al., 2024; Grattafiori et al., 2024; Olmo et al., 2026; Bjorck et al., 2025). This practice implicitly assumes that improvements to the base model will carry over after post-training. Recent work has shown that this assumption can fail: beyond a certain point, extending pretraining improves pretraining loss while degrading performance after post-training (Springer et al., 2025). This motivates the central question of our work: can pretraining optimization choices yield better models after post-training even when they do not improve the base model itself?

The crux of the gap between pretraining and post-training performance is that base-model evaluation ignores a key property: how stable a model's capabilities are under the parameter updates introduced by post-training. Models that are sensitive to these updates "forget" pretrained abilities (Goodfellow et al., 2013; Kirkpatrick et al., 2017), regardless of how strong they look as base models. This points toward optimization choices that minimize not just pretraining loss, but also sensitivity to post-training-induced parameter perturbations.

We study three pretraining interventions targeting this sensitivity. First, we evaluate Sharpness-Aware Minimization (SAM) (Foret et al., 2021), which explicitly penalizes loss curvature and may therefore reduce sensitivity to post-training-induced parameter changes. Second, we study two simpler alternatives: increasing the peak learning rate and shortening the learning-rate annealing period at the end of training, both motivated by prior work relating learning rate to loss curvature (Cohen et al., 2021; Damian et al., 2023).

---

[*]Equal contribution [†]Equal advising [1]Carnegie Mellon University. Correspondence to: Ishaan Watts <iwatts@cs.cmu.edu>, Catherine Li <catheri4@andrew.cmu.edu>.

*Proceedings of the $43^{rd}$ International Conference on Machine Learning*, Seoul, South Korea. PMLR 306, 2026. Copyright 2026 by the author(s).

Across over 80 pretraining runs and 3,500 fine-tuning experiments, we find that each intervention yields a base model with a superior "learning–forgetting tradeoff"—e.g., SAM produces 80% less forgetting on StarCoder at matched fine-tuning loss—and that the advantage grows with token budget.

Our controlled experiments identify the learning-rate annealing phase as a critical determinant of the learning–forgetting tradeoff, motivating sharpness-aware updates during the late stages of training. To validate this at scale, we mid-train OLMo-2-1B (OLMo et al., 2024) on 50B tokens with SAM—a late-stage intervention on top of a fully pretrained 4T-token checkpoint—and then post-train on four standard datasets. Compared to mid-training with the standard OLMo-2 recipe, **SAM yields 31% less forgetting after MetaMath post-training** and **40% less forgetting under 4-bit quantization** (bitsandbytes NF4; Figure 1, left and right respectively). Because these gains come from applying SAM over only a small fraction of total training compute, practitioners can capture much of its benefit by deploying it selectively during late training rather than throughout pretraining.

These results identify parameter sensitivity to post-training updates as the critical—and overlooked—link between pretraining dynamics and downstream performance, and motivate selecting pretraining recipes for both base-model quality and low sensitivity to downstream parameter shifts.

## 2. Preliminaries

Canonically, optimization choices are made to minimize pretraining loss. In this work, we focus on understanding how design choices during pretraining affect the *downstream model*, after some sort of modification such as fine-tuning or quantization.

### 2.1. Downstream properties of the pretrained model

Let $\theta_{\mathrm{PT}}$ denote the pretrained model. We study the following downstream properties of the pretrained model.

**The learning-forgetting tradeoff in fine-tuning.** Pretrained models are typically fine-tuned on task-specific or domain-specific data to introduce specific capabilities. However, recent works have documented that optimizing for a specific task often leads to the forgetting of pretrained capabilities (Goodfellow et al., 2013; Kirkpatrick et al., 2017; Springer et al., 2025). When a base model $\theta_{\mathrm{PT}}$ is fine-tuned to obtain $\theta_{\mathrm{FT}}$, we measure the "learning" effect via the validation loss on the fine-tuning data, $\mathcal{L}_{\mathrm{FT}}(\theta_{\mathrm{FT}})$. We measure the induced "forgetting" effect by evaluating the pretraining loss on the *fine-tuned* weights, $\mathcal{L}_{\mathrm{PT}}(\theta_{\mathrm{FT}})$. Since this balance is sensitive to optimization choices (e.g., choices of hyperparameters), a single fine-tuned checkpoint does not provide a complete picture of the adaptation capability of the pretrained model. Therefore, to characterize the degree to which $\theta_{\mathrm{PT}}$ can be adapted via fine-tuning, we analyze the *learning-forgetting tradeoff*, defined as the set:

$$\left\{ \left( \mathcal{L}_{\mathrm{PT}}(\theta_{\mathrm{FT}}), \mathcal{L}_{\mathrm{FT}}(\theta_{\mathrm{FT}}) \right) \mid \theta_{\mathrm{FT}} \in \Theta_{\mathrm{FT}}(\theta_{\mathrm{PT}}) \right\},$$

where $\Theta_{\mathrm{FT}}(\theta_{\mathrm{PT}})$ represents the set of all models obtained by fine-tuning $\theta_{\mathrm{PT}}$ under varying fine-tuning configurations. We are interested, primarily, in the Pareto frontier of this set, which characterizes, under optimal fine-tuning configurations, how well the base model can learn a downstream task without compromising too much of its pretrained capability.

**The compression-forgetting tradeoff via quantization.** Beyond fine-tuning, it is often desirable to reduce the serving cost of the model at inference time by compressing the model to enable more efficient use of the GPU. Compression inevitably leads to a degradation in model capabilities, and we characterize the tradeoff between compression and forgetting by tracking the base model performance vs. the degree of compression, analogous to the learning-forgetting tradeoff. In this work, we study quantization, a popular and effective compression approach. We leave alternative methods of compression, such as model weight pruning to future work. We characterize the compression-degradation tradeoff analogously to the learning-forgetting tradeoff, by tracking the compression rate (e.g., number of bits per parameter) against the resulting pretraining loss.

### 2.2. Sharpness as a local approximation for forgetting

Our core intuition is that fine-tuning lands the parameters near the base model, and thus can be thought of as a local perturbation. Training the model to limit sensitivity to local perturbations, intuitively, should enable fine-tuning without forgetting. This leaves open the question, how should we optimize to limit the model sensitivity to local perturbations? A natural candidate is to reduce the *directional curvature* of the loss landscape along directions relevant for fine-tuning. Precisely, the directional curvature of the loss landscape with Hessian $H$ along the direction of a given vector $u$ is the quantity:

$$\kappa(u; H) = \frac{1}{\|u\|^2} u^\top H u. \tag{1}$$

Our intuition is illustrated by examining a local approximation of the pretraining loss around the base model parameters $\theta_{\mathrm{PT}}$. Consider the perturbation $\theta_{\mathrm{PT}} + \Delta$ that arises as a result of a downstream modification, such as fine-tuning or quantization. Letting $\mathcal{L}$ denote the pretraining loss $\mathcal{L}_{\mathrm{PT}}$ and $\theta$ the pretraining parameters $\theta_{\mathrm{PT}}$, a second-order Taylor expansion yields:

$$\mathcal{L}(\theta + \Delta) \approx \mathcal{L}(\theta) + \nabla \mathcal{L}(\theta)^\top \Delta + \tfrac{1}{2} \Delta^\top H \Delta, \tag{2}$$

where $H := \nabla^2 \mathcal{L}(\theta)$ is the Hessian of the pretraining loss. We typically pretrain for a large number of steps, making

the gradient small, and thus the increase in loss (forgetting) is dominated by the quadratic term, where $\Delta_{\text{FT}}$ denotes the fine-tuning direction:

$$\mathcal{L}_{\text{PT}}(\theta_{\text{FT}}) - \mathcal{L}_{\text{PT}}(\theta_{\text{PT}}) \approx \tfrac{1}{2}\Delta_{\text{FT}}^{\top}H\Delta_{\text{FT}} \tag{3}$$

$$= \tfrac{1}{2}\|\Delta_{\text{FT}}\|^2\kappa(\Delta_{\text{FT}}; H) \tag{4}$$

This approximation indicates that, for small perturbations, forgetting is determined by two factors: the *distance moved* $\|\Delta_{\text{FT}}\|^2$ and the *curvature in the direction of fine-tuning* $\kappa(\Delta_{\text{FT}}; H)$. High curvature in this direction (henceforth *sharpness*) leads to significant forgetting, whereas low curvature allows parameter adaptation with minimal impact on the original task performance. Therefore, reducing the curvature in the direction of fine-tuning may limit forgetting.

**Caveats with curvature.** Optimizing to reduce curvature in the direction of fine-tuning is potentially impractical. For one, at pretraining time, we do not assume knowledge of the fine-tuning task; the pretraining methodology should be agnostic of the downstream task. Second, even if the downstream task is known, the direction of fine-tuning for the final checkpoint may be unknown during training. Third, minimizing curvature along perturbation directions does not provide any theoretical guarantee of low sensitivity to perturbations larger than the radius around which the pretraining loss is well-approximated by a quadratic. Nonetheless, we demonstrate that two optimization recipes for minimizing curvature along two general directions, which we discuss below can lead to limited sensitivity in directions and perturbation distances relevant for fine-tuning in Section 3. In addition, we revisit these caveats to check whether our recipes do in fact reduce curvature along fine-tuning-relevant directions, and whether this curvature reduction is sufficient to explain improved robustness to forgetting in Appendix F.

## 2.3. Optimization recipes

Motivated by the connection between the curvature of the loss landscape and forgetting, we consider two distinct optimization mechanisms for inducing small loss curvature along certain directions in parameter space: an *explicit* approach via Sharpness-Aware Minimization, and an *implicit* approach via learning rate dynamics.

### 2.3.1. SHARPNESS-AWARE MINIMIZATION (SAM)

SAM (Foret et al., 2021) explicitly searches for minima that remain low-loss under parameter perturbations within a specified neighborhood. Given the pretraining objective $\mathcal{L}_{\text{PT}}(\theta)$ and a radius $\rho > 0$, SAM solves the robust optimization problem:

$$\min_{\theta} \max_{\|\epsilon\|_2 \leq \rho} \mathcal{L}_{\text{PT}}(\theta + \epsilon). \tag{5}$$

In practice, this is approximated by a first ascent step in the direction of the gradient to find a perturbed weight vector

and then updating the original parameters $\theta$ to take a descent step using the gradient evaluated at that perturbed location. SAM with a batch size of 1 is thought to minimize the trace of the Hessian, $\text{Tr}(H) = \sum_i \lambda_i(H)$ (curvature in the "average" direction), while full-batch SAM tracks the worst-case directional curvature $\lambda_{\max}(H)$ (Wen et al., 2023b); we train with a small batch size and so intuitively expect our SAM updates to approximately reduce $\text{Tr}(H)$.

### 2.3.2. LEARNING RATES AND THE EDGE OF STABILITY

Learning rate is thought to implicitly regularize curvature via the "Edge of Stability" phenomenon (Cohen et al., 2021). This phenomenon posits that gradient descent drives the maximum eigenvalue of the Hessian $\lambda_{\max}(H)$ to be implicitly capped by the learning rate $\eta$, with dynamics hovering at $\lambda_{\max}(H) \approx 2/\eta$. This suggests that the pretraining learning rate may play an important role for post-training sensitivity, motivating our investigation into the learning rate and its annealing schedule.

## 3. Experiments

We now evaluate the sharpness-minimizing recipes from Section 2.3, asking: (1) whether explicit and implicit sharpness minimization improve the learning–forgetting tradeoff (Sections 3.2–3.3); (2) whether this robustness extends beyond fine-tuning to quantization and weight perturbations (Section 3.4); and (3) whether these recipes are practical at scale (Section 3.5).

### 3.1. Experimental setup

**Pretraining.** We pretrain OLMo-style models (Groeneveld et al., 2024) at 20M, 60M, and 150M parameters on 4B–192B DCLM-Baseline tokens (Li et al., 2024), comparing AdamW (Loshchilov & Hutter, 2019) and SAM (Foret et al., 2021) with cosine (Loshchilov & Hutter, 2017) and WSD learning rate schedules (Hu et al., 2024). AdamW learning rates are tuned for pretraining validation loss and reused elsewhere, which favors AdamW; for SAM, we choose $\rho = 0.05$ from small-scale tuning (Appendix C.1). Unless stated otherwise, "AdamW" denotes the standard recipe used in OLMo-2, OLMo-3, and LLaMA-3: AdamW with cosine annealing and hyperparameters selected to minimize pretraining loss (OLMo et al., 2024; Olmo et al., 2026; Grattafiori et al., 2024).

**Fine-tuning.** We evaluate the learning-forgetting tradeoff for five publicly available datasets: StarCoder (Li et al., 2023) (code generation), GSM8K (Cobbe et al., 2021) and StackMathQA (Zhang, 2024) (mathematical reasoning), Tülu-3 (Lambert et al., 2025) (instruction following), and MusicPile (Yuan et al., 2024) (domain-specific). Fine-tuning uses AdamW with a cosine schedule, learning rates

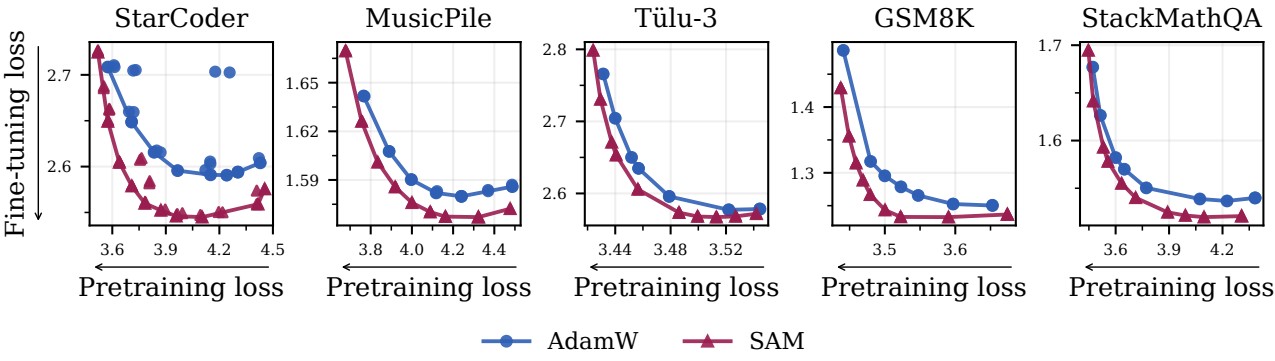

*Figure 2.* **SAM consistently yields pretrained checkpoints that forget less when fine-tuned to the same performance as AdamW counterparts.** We pretrain OLMo-60M models with a cosine schedule using AdamW and SAM on 192B tokens and fine-tune on five datasets. SAM achieves a better learning-forgetting frontier.

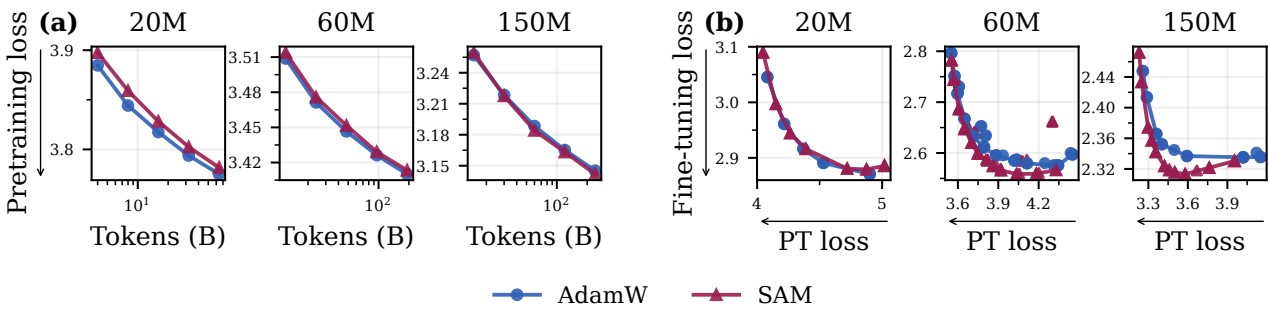

*Figure 3.* **Comparison of SAM and AdamW with model size. (a)** Across model sizes and token budgets, SAM-pretrained models achieve a worse or similar pretraining loss compared to AdamW. However, better pretraining loss alone does not translate into mitigating forgetting. **(b)** We pretrain OLMo models of sizes 20M, 60M, and 150M on similar *token-per-parameter* ratios (800) and then fine-tune on StarCoder. We observe that the improvements of SAM over AdamW do not diminish, or in some cases even improve, with scaling model size.

from $1 \times 10^{-6}$ to $1 \times 10^{-2}$, batch size 64, and no weight decay. Each run lasts one epoch or 10M tokens, whichever comes first (Appendix C.2).

**Evaluating the learning- and compression-forgetting tradeoffs.** We evaluate each fine-tuned checkpoint for fine-tuning loss and pretraining loss on fixed fine-tuning and pretraining validation datasets, respectively, with the pretraining validation loss used as our metric for forgetting (in contrast to the benchmark-suite drop reported at 1B in Section 3.5.1). Taken as a set (defined in Section 2.1), these points estimate the learning-forgetting Pareto frontier. For the compression-forgetting tradeoff we compare the pretraining validation loss of the base models at full bit-width precision (bf16) against the same models quantized to 4 bits.

**Evaluating sensitivity to Gaussian perturbations.** Many post-training and inference-time methods perturb weights directly, including fact updating (De Cao et al., 2021) and concept editing (Wang et al., 2024a). Rather than evalu-

ating all possible model updates, we use a task-agnostic proxy: isotropic Gaussian noise added to the pretrained weights, with sensitivity measured by pretraining validation loss. More details in Appendix C.3.

### 3.2. Explicitly minimizing sharpness mitigates forgetting

**Main learning-forgetting result in the token-matched setting.** Across 20M–150M parameters, 4B–192B tokens, and five fine-tuning datasets, SAM gives better learning–forgetting frontiers than AdamW. In the canonical OLMo-60M, 192B-token setting (Figure 2), **SAM reduces Star-Coder forgetting by 80% at matched fine-tuning loss**, with degradation of $+0.1$ instead of $+0.5$. Gains are largest for StarCoder and MusicPile and smallest for Tülu-3, likely because Tülu-3 is closer to DCLM. SAM improves fine-tuned models despite similar or worse base-model loss (Figure 3.a). Results for the 20M and 150M models are reported in Appendix E.1.1.

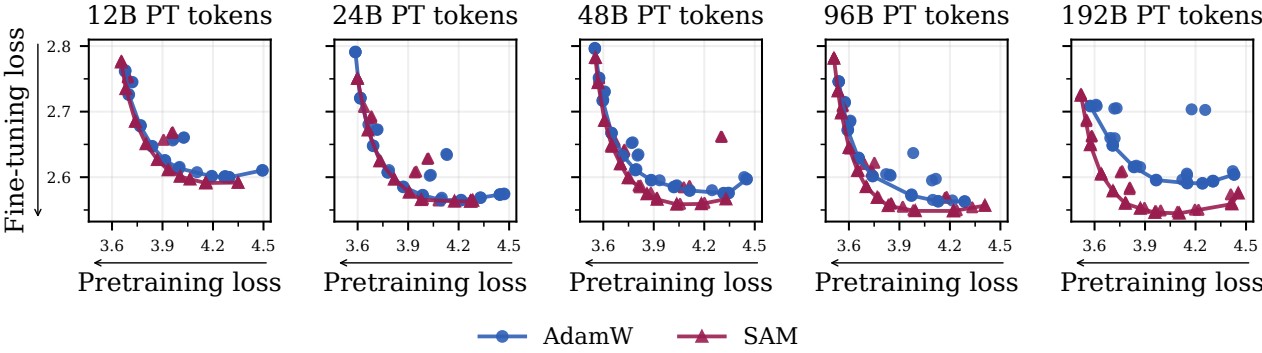

*Figure 4.* **SAM's improvement over AdamW grows with scaling pretraining tokens.** We pretrain OLMo-60M models with a cosine schedule using AdamW and SAM on 4B to 192B tokens and fine-tune on StarCoder. The gap between SAM and AdamW widens as we scale pretraining tokens.

**SAM improves the LF-tradeoff gap as the token budget scales.** For 60M models fine-tuned on StarCoder, the SAM–AdamW gap widens from 12B to 192B pretraining tokens (Figure 4). AdamW checkpoints become increasingly sensitive with more training, matching Springer et al. (2025); SAM mitigates this, with its strongest advantage over AdamW in the high-token regime. The trend holds across our full sweep of model sizes and fine-tuning datasets (see Appendix E.1.2).

**The improvement of SAM persists over model scale.** At a fixed *token-per-parameter* ratio, SAM continues to outperform AdamW as model size increases. The gap is larger at 60M and 150M than at 20M, suggesting that SAM's benefit may grow with scale, especially on MusicPile and Stack-MathQA. We observe similar trends across other datasets (Appendix E.1.3).

**Pretraining loss-matched setting.** One possible explanation is that SAM trains more slowly, acting like early stopping before AdamW reaches sensitive regimes (Springer et al., 2025). To rule this out, we compare OLMo-60M checkpoints at matched base pretraining loss (details in Appendix C.4): for each, we pick a target fine-tuning loss and report the least forgetting among runs that reach it (Figure 5). SAM consistently forgets less at the same pretraining loss. On StarCoder, MusicPile, and StackMathQA, AdamW becomes more fragile as pretraining loss falls to around 3.43, while SAM checkpoints continue to show no such sensitivity and improve monotonically in forgetting with pretraining, up to the budgets we consider. Similar trends for the 20M and 150M models are reported in Appendix E.1.4.

**SAM's gains stack with continual learning methods.** We further find that SAM's benefits compound with explicit continual learning techniques: combining SAM-pretrained checkpoints with EWC (Kirkpatrick et al., 2017) during fine-tuning yields a strictly better learning–forgetting frontier than EWC applied on top of AdamW (Appendix E.1.5).

**Summary:** SAM dominates AdamW on the learning–forgetting tradeoff (e.g., 80% less forgetting on StarCoder at matched fine-tuning loss). The gap widens with token budget and persists at matched pretraining loss.

### 3.3. Implicitly minimizing base model sharpness mitigates forgetting

We turn to investigating how simply setting the *learning rate*—with no explicit sharpness penalty—can influence the learning-forgetting and compression-forgetting tradeoffs. As we discuss in Section 2.3.2, increasing the learning rate has been shown to implicitly penalize base model sharpness via the Edge-of-Stability mechanism. In this section we investigate two aspects of the learning rate: (1) the maximum "peak" learning rate, and (2) the duration of learning rate annealing, both of which we find to influence the learning-forgetting tradeoff.

**Higher peak learning rates during pretraining improve the learning-forgetting tradeoff.** We sweep the peak AdamW learning rate under cosine pretraining schedule, then fine-tune on StarCoder. Larger peak learning rates consistently improve the learning–forgetting frontier (Figure 6.b), even though base model pretraining loss is minimized at a moderate learning rate (Figure 6.a, optimum marked with an asterisk). The pattern is robust across schedules: both cosine and WSD (Appendix E.2) show the same monotonic relationship between peak learning rate and the learning–forgetting frontier, even when base-model pretraining loss begins to degrade.

**Shorter annealing periods improve the learning–forgetting tradeoff.** We vary the WSD decay length $d$:

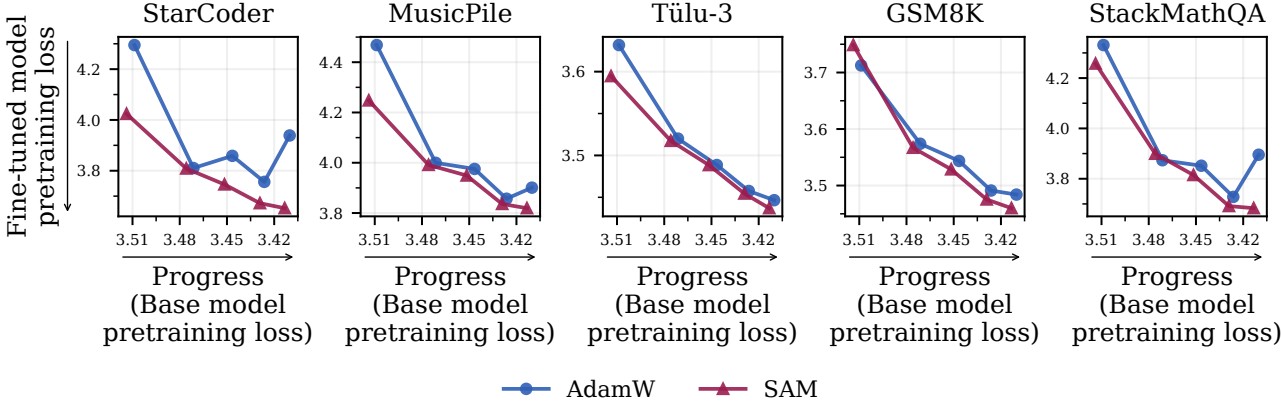

*Figure 5.* **SAM delays the onset of catastrophic overtraining.** We pretrain OLMo-60M models with a cosine learning rate schedule using AdamW and SAM and then fine-tune on five datasets. We plot the minimum achievable pretraining loss such that the fine-tuning loss is below a threshold (more details in Appendix C.4) as a function of the base model loss. Once the AdamW-trained models reach a certain base model pretraining loss, the minimum achievable pretraining loss after fine-tuning often begins to increase with further improvement. In contrast, SAM-trained models continue to exhibit stable or improving tradeoffs over the same regime.

after warmup, the learning rate stays fixed for $N - d$ steps, then decays to 10% of its peak over the final $d$ steps. For 60M models trained for 192B tokens and fine-tuned on Star-Coder, shorter annealing gives consistently better frontiers (Figure 7.b). The best setting for the learning–forgetting tradeoff is the shortest schedule we tried, 5% of training, while a duration of 20% was optimal (Figure 7.a) when minimizing base model pretraining loss alone. We observe similar trends across other datasets (see Appendix E.3).

> **Summary:** Higher peak learning rates and shorter annealing periods both improve the learning–forgetting tradeoff, even when they hurt pretraining loss.

### 3.4. Beyond fine-tuning

Forgetting can happen without fine-tuning. To test whether optimization choices affect this broader form of sensitivity, we first study the compression–forgetting tradeoff via quantization (described in Section 3.1). We then consider a more generic perturbation: Gaussian noise added directly to the weights. Unlike quantization, this probe is not tied to a specific deployment method and instead measures the model's average-case robustness to small local parameter changes around the pretrained checkpoint.

**SAM improves the compression–forgetting tradeoff.** For OLMo-60M models trained on 12B–192B tokens, SAM consistently lowers the pretraining-loss increase from 4-bit quantization (Figure 8.a). Quantization becomes much more damaging at high token budgets, as also observed by Kumar et al. (2025). **In this high-token budget regime, SAM yields roughly 2–3× less quantization-induced loss**

**increase than the baseline.** At a lower budget of 24B tokens, SAM lowers degradation from 0.14 to 0.08, a 42% reduction relative to AdamW at the same budget. Similar trends are reported for 20M and 150M models in Appendix E.1.6.

**SAM-trained checkpoints are broadly less sensitive to perturbations.** The same pattern holds under isotropic Gaussian weight noise (Figure 8.b). SAM checkpoints suffer less pretraining-loss degradation than AdamW checkpoints, again with the largest gap at the highest token budgets we evaluate. Similar trends are reported for 20M and 150M models in Appendix E.1.7.

**Higher peak learning rates and shorter annealing periods improve the compression–forgetting tradeoff and sensitivity to perturbations.** Our observations for SAM are mirrored when considering higher peak learning rates and shorter annealing periods. Increasing the peak learning rate reduces 4-bit quantization degradation (Figure 6.c), even at $3 \times 10^{-3}$, the largest rate tried and $10\times$ the base-model-optimal value. Shortening WSD annealing also helps: 10% annealing beats the base-model-optimal 20% (Figure 7.c).

> **Summary:** Sharpness-aware pretraining recipes reduce forgetting under 4-bit quantization and Gaussian weight perturbations (under quantization, SAM yields up to a 2–3× improvement in high-token regimes).

### 3.5. A scalable recipe for sharpness minimization

We have shown that sharpness minimization during pretraining improves both learning– and compression– forgetting tradeoffs. However, SAM roughly doubles per-step compute

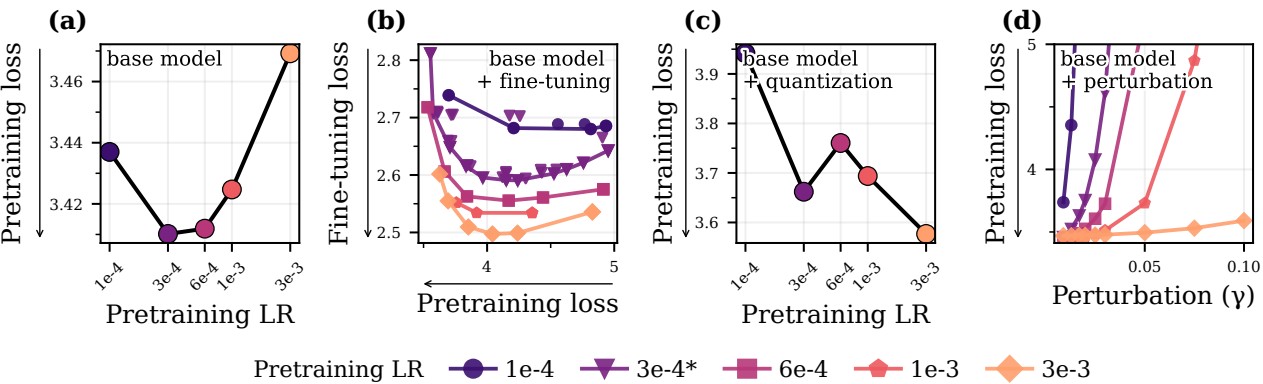

*Figure 6.* **Higher peak learning rates improve the learning-forgetting tradeoff.** We vary the peak pretraining learning rate for 60M models with a cosine schedule on 192B tokens. **(a)** Pretraining loss vs. peak learning rate. **(b)** Learning-forgetting Pareto frontier on StarCoder. The asterisk in the legend marks the peak learning rate that achieves the lowest base-model pretraining loss. **(c)** 4-bit quantized pretraining loss vs. peak learning rate. **(d)** Perturbed pretraining loss vs. perturbation magnitude $\gamma$.

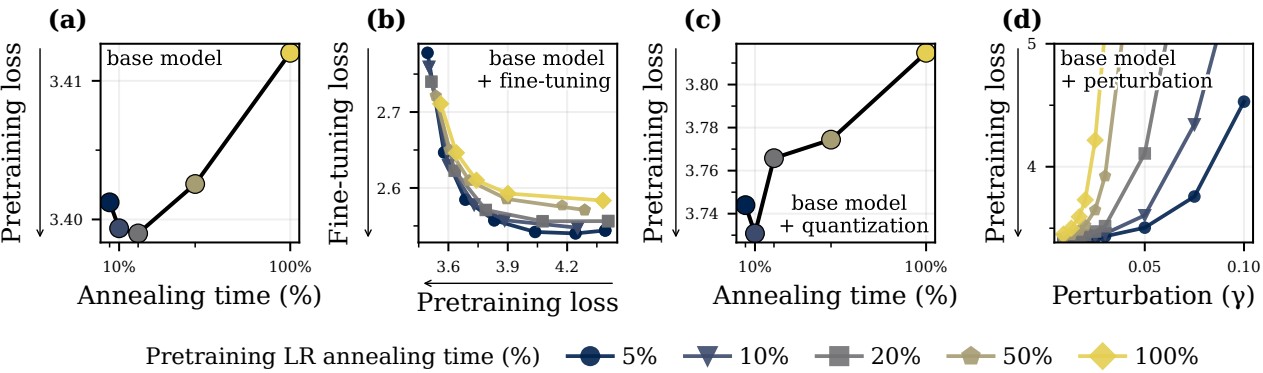

*Figure 7.* **Shorter annealing periods improve the learning-forgetting tradeoff.** We vary the annealing duration as a percentage of total training steps for 60M models with a WSD schedule. **(a)** Pretraining loss vs. anneal percent. **(b)** Learning-forgetting Pareto frontier on StarCoder. **(c)** 4-bit quantized pretraining loss vs. anneal percent. **(d)** Perturbed pretraining loss vs. perturbation magnitude $\gamma$.

relative to AdamW, making full-pretraining SAM expensive at scale. Motivated by Section 3.3, where we identify annealing as the critical phase for the learning–forgetting tradeoff, we apply SAM only during learning-rate annealing. This concentrates sharpness-aware updates where they matter most, yielding a practical path to sharpness minimization at pretraining scale.

**Experimental setup.** We use a WSD schedule with AdamW during warmup and the constant phase, then switch to SAM for the learning rate decay phase (10% of training) using the hyperparameters from Section 3.1. We pretrain a 60M OLMo model for 192B tokens and evaluate StarCoder fine-tuning (other datasets in Appendix E.4) and 4-bit quantization against WSD with AdamW throughout.

**Annealing with SAM improves the learning–forgetting**

**tradeoff.** On StarCoder, switching to SAM only during annealing strictly improves the baseline frontier (Figure 9.a), giving lower fine-tuning loss at the same forgetting and less forgetting at the same fine-tuning loss. This recovers much of full-run SAM's benefit at a fraction of the compute cost.

**Annealing with SAM improves the compression– forgetting tradeoff.** The same late-SAM recipe reduces 4-bit quantization degradation (Figure 9.b), with the largest gains at high token budgets where AdamW becomes most compression-sensitive.

**Summary:** Applying SAM only during WSD decay improves fine-tuning and quantization robustness while adding roughly the annealing fraction in compute: ∼10% here versus ∼100% for full-run SAM.

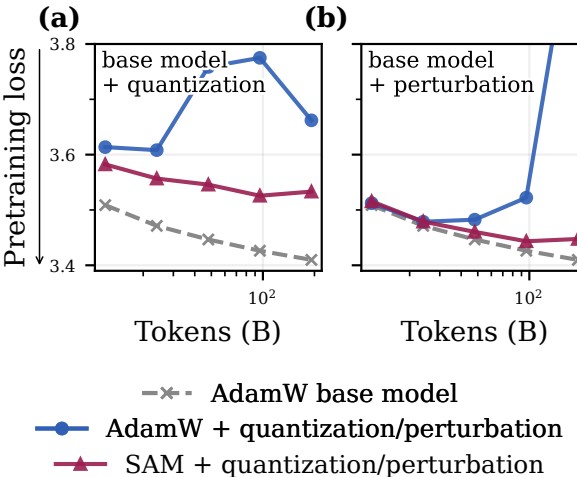

Figure 8. **SAM improves sensitivity to post-training quantization and Gaussian perturbations.** We pretrain OLMo-60M for budgets ranging from 12B to 192B tokens. **(a)** 4-bit quantized pretraining loss vs. pretraining tokens, with the unquantized AdamW reference for scale. **(b)** Perturbed pretraining loss at $\gamma = 0.025$ vs. pretraining tokens.

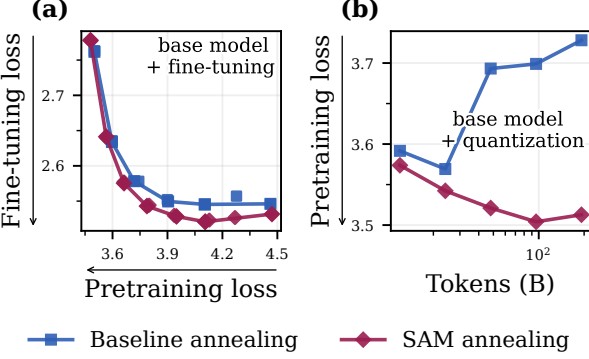

Figure 9. **Annealing with SAM improves downstream performance over baseline annealing.** We pretrain OLMo-60M for 192B tokens with a WSD schedule and a 10% anneal, comparing AdamW throughout (*baseline annealing*) to a recipe that switches to SAM during the decay phase. **(a)** Learning-forgetting Pareto frontier after fine-tuning on StarCoder (10M tokens). **(b)** 4-bit quantized pretraining loss vs. pretraining tokens (12B–192B).

### 3.5.1. APPLYING SHARPNESS-AWARE ANNEALING AT SCALE

We next ask whether sharpness-aware annealing remains effective at 1B scale.

**Experimental setup.** Starting from the fully pretrained OLMo-2-1B checkpoint (OLMo et al., 2024), we mid-train for 50B tokens on the Dolmino mixture using the original OLMo-2-1B linear-annealing recipe, which we refer to as *OLMo baseline*. Our SAM run changes only the optimizer

during this mid-training phase, using $\rho = 0.05$ and the hyperparameters in Appendix B.1. We evaluate forgetting by post-training both checkpoints on MetaMath (Yu et al., 2023), StackMathQA (Zhang, 2024), Tülu-3 (Lambert et al., 2025), and MusicPile (Yuan et al., 2024), choosing hyperparameters that match fine-tuning loss so that forgetting is compared at fixed downstream learning (Appendix B.2). We measure degradation as the drop in average OLMo pretraining benchmark-suite (refer to Appendix B.1.2) performance before and after post-training, and separately evaluate compression robustness by applying 4-bit `bitsandbytes` quantization (Dettmers et al., 2023). Since the mid-trained base checkpoints are nearly tied—43.2 for the OLMo baseline and 42.9 for SAM—we report all degradation relative to 43.2, slightly favoring the baseline.

**Sharpness-aware mid-training mitigates forgetting despite a weaker base model.** SAM mid-training reduces catastrophic forgetting by 31% on MetaMath (Figure 1), 22% on StackMathQA, and 35% on Tülu-3, despite slightly worse base performance. Thus, the stronger base checkpoint is not necessarily the stronger post-trained checkpoint: SAM produces a model that is slightly worse before post-training, but substantially more robust after post-training across all four datasets. The learning–forgetting Pareto frontiers can be seen in Appendix D.2.

**Sharpness-aware mid-training improves post-training quantization.** The same robustness extends beyond fine-tuning. Under 4-bit quantization, the SAM mid-trained model loses 40% less benchmark performance than the OLMo baseline, showing that sharpness-aware mid-training also improves compression robustness at scale. Detailed evaluation results in Appendix D.1.

## 4. Related work

**Catastrophic forgetting and continual learning.** Catastrophic forgetting—the failure of neural networks to retain prior knowledge while learning new information—is a central challenge in deep learning (French, 1999; Goodfellow et al., 2013). Continual learning methods address this problem in explicit task sequences through regularization (Kirkpatrick et al., 2017; Zenke et al., 2017; Aljundi et al., 2018), replay (Shin et al., 2017), gradient projection (Lopez-Paz & Ranzato, 2017; Chaudhry et al., 2018), and architectural expansion (Rusu et al., 2016; Zhou et al., 2023). Our learning–forgetting tradeoff is an instance of the classical stability–plasticity dilemma, with fine-tuning loss measuring plasticity and pretraining-loss preservation measuring stability (Lopez-Paz & Ranzato, 2017). Unlike standard continual learning, we study forgetting as a property of the pretrained checkpoint itself, making our intervention—modifying the pretraining process—orthogonal to fine-tuning-time methods.

**Loss of plasticity and catastrophic overtraining.** A closely related phenomenon is *loss of plasticity*: continued optimization can make networks harder to adapt even as training loss improves. This effect appears in RL as *primacy bias* (Nikishin et al., 2022), in supervised learning via warm-starting (Ash & Adams, 2020), and in large-scale pretraining (Mehta et al., 2023), where long optimization horizons expose plasticity failures (Dohare et al., 2024) driven by changes in network geometry (Tang et al., 2025). Recent work addresses related challenges through curriculum and mid-training strategies (Gururangan et al., 2020; Liu et al., 2026; Wang et al., 2025; Kotha & Liang, 2026) or prompting-based mechanisms (Kotha et al., 2024). Yet core pretraining choices—including learning-rate scaling (Bjorck et al., 2025), token budgets (Grattafiori et al., 2024), and optimizer selection (Jordan et al., 2024)—are still typically selected by pretraining loss, even though lower pretraining loss need not imply better downstream performance (Liu et al., 2022). Recent work shows that over-optimizing pretrained models can actively degrade fine-tuning performance (Springer et al., 2025). We study this *catastrophic overtraining* problem and propose pretraining-time interventions that improve downstream robustness.

**Sharpness and generalization.** We connect this loss of plasticity to the geometry of the loss landscape. The relationship between flat minima and generalization has been widely studied (Keskar et al., 2016), though sharpness must be interpreted carefully because it can depend on parameterization (Dinh et al., 2017). Despite ongoing debate about the exact relationship (Andriushchenko et al., 2023a), sharpness remains a useful predictor of generalization (Jiang* et al., 2020), and wider minima can be encouraged both by standard SGD dynamics and by explicit optimization interventions (Baldassi et al., 2020). In our setting, the relevant notion of generalization is not only test-set accuracy, but *robustness to later modification*: a useful pretrained checkpoint should remain performant after fine-tuning, quantization, or other downstream perturbations.

**Training to minimize sharpness.** Several optimization methods explicitly reduce sharpness by minimizing sensitivity to local parameter perturbations. Entropy-SGD (Chaudhari et al., 2016) and Sharpness-Aware Minimization (SAM) (Foret et al., 2021) cast training as robust optimization over local neighborhoods, and many efficient SAM variants have since been proposed (Kwon et al., 2021; Zhuang et al., 2022; Du et al., 2023). Sharpness-aware training has also been used to improve continual learning (Bian et al., 2024). Although theory continues to clarify which notions of sharpness SAM actually minimizes (Wen et al., 2023b), and sharpness is only one component of generalization (Wen et al., 2023a; Springer et al., 2024), these methods provide a direct mechanism for smoothing the landscape. We use this mechanism to target a different objective: reducing forgetting after downstream modification.

**Implicit regularization via training dynamics.** Sharpness can also be shaped implicitly by standard optimization hyperparameters. Early in training, large learning rates delay memorization and improve generalization (Li et al., 2019); large step sizes can act like label noise, promoting sparser and more generalizable features (Andriushchenko et al., 2023b); and large-learning-rate dynamics can drive training toward the edge of stability, where the maximum Hessian eigenvalue is controlled by the step size (Cohen et al., 2021). Later-stage dynamics are equally important: learning-rate schedules, especially annealing, can determine the final sharpness of the solution (Zhou et al., 2025) and affect post-training outcomes such as PTQ (Catalan-Tatjer et al., 2026).

## 5. Conclusion

In this work, we asked whether the pretraining recipe that produces the best base model is also the recipe that produces the best model after post-training. Across controlled experiments, we found that this need not be the case: explicit sharpness minimization with SAM, as well as implicit sharpness control through larger peak learning rates and shorter annealing periods, improved the learning–forgetting and compression–forgetting tradeoffs despite not always improving base-model pretraining loss. Our Hessian analysis connected these gains to reduced fine-tuning-directional sharpness, and our OLMo-2-1B experiments showed that a short SAM mid-training phase can reduce forgetting after post-training and 4-bit quantization at scale.

These results also expose several open questions. At 1B scale, SAM mid-training improved post-training robustness on MetaMath, StackMathQA, and Tülu-3, but did not improve forgetting after post-training on MusicPile; understanding which properties of downstream data determine when sharpness reduction helps is an important direction for future work. More generally, we studied supervised fine-tuning and post-training quantization, leaving open how checkpoint sharpness affects other post-training regimes, including reinforcement learning, preference optimization, adapters, and alternatives to SFT. Finally, although SAM, larger peak learning rates, and shorter annealing each reduced useful notions of sharpness, they are still proxies: pretraining loss alone did not capture adaptability, and a central open problem is to identify objectives or validation criteria that more directly predict post-training robustness.

The broader lesson is that pretraining should be optimized for the full model-development pipeline, not for the base checkpoint in isolation: optimal hyperparameters should be predicted for the post-trained model, and adaptability treated as a first-class criterion alongside pretraining loss.

## Acknowledgements

We gratefully acknowledge support from Apple, Google, Jane Street, the National Science Foundation and the FLAME cluster at Carnegie Mellon University.

This material is based upon work supported by the National Science Foundation Graduate Research Fellowship under Grant No. DGE2140739. Any opinions, findings, conclusions, or recommendations expressed in this material are those of the authors and do not necessarily reflect the views of the National Science Foundation.

The authors thank Christina Baek, Gaurav Ghosal, Ziqian Zhong, and Lawrence Feng for helpful discussions.

## Impact Statement

Our work provides insights into how pre-training optimization affects post-training stability in large language models. By understanding the connection between loss landscape sharpness and catastrophic forgetting, researchers and practitioners can design models that are more robust to post-train modifications, supporting continual learning. More stable post-training models also enable reliable quantization, allowing for faster inference without sacrificing performance. Although we do not target a specific application, these insights can benefit the broader community by reducing computational costs, lowering environmental impact, and facilitating the development of robust and long-lived models.

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

# A. Definitions

## A.1. Optimizers

### A.1.1. ADAMW

AdamW (Loshchilov & Hutter, 2019) is an adaptive gradient-based optimizer that combines the benefits of momentum and per-parameter learning rate adaptation, while decoupling weight decay from the gradient update. This decoupling corrects a flaw in standard Adam and leads to improved generalization.

Let $g_t = \nabla_\theta \mathcal{L}(\theta_t)$ denote the gradient at step $t$. AdamW maintains exponential moving averages of the first and second moments:

$$m_t = \beta_1 m_{t-1} + (1 - \beta_1) g_t, \quad v_t = \beta_2 v_{t-1} + (1 - \beta_2) g_t^2.$$

After bias correction,

$$\hat{m}_t = \frac{m_t}{1 - \beta_1^t}, \quad \hat{v}_t = \frac{v_t}{1 - \beta_2^t}.$$

The parameter update is given by

$$\theta_{t+1} = \theta_t - \alpha \frac{\hat{m}_t}{\sqrt{\hat{v}_t} + \epsilon} - \alpha \lambda \theta_t,$$

where $\alpha$ is the learning rate and $\lambda$ is the weight decay coefficient.

### A.1.2. SHARPNESS-AWARE MINIMIZATION

Sharpness-Aware Minimization (SAM) (Foret et al., 2021) is an optimization framework designed to improve generalization by explicitly favoring flatter minima. Unlike AdamW, which updates parameters based on gradients at the current point, SAM seeks parameters whose neighborhood exhibits uniformly low loss. This results in solutions that are less sensitive to small perturbations and empirically generalize better.

At each step, SAM first computes a perturbation in the direction of the gradient:

$$\epsilon_t = \rho \frac{\nabla_\theta \mathcal{L}(\theta_t)}{\|\nabla_\theta \mathcal{L}(\theta_t)\|_2},$$

where $\rho$ controls the size of the neighborhood. The parameters are temporarily perturbed to $\tilde{\theta}_t = \theta_t + \epsilon_t$, and the final update is computed using the gradient at this perturbed point:

$$\theta_{t+1} = \theta_t - \alpha \nabla_\theta \mathcal{L}(\tilde{\theta}_t).$$

By optimizing for robustness within a local neighborhood, SAM encourages convergence to flatter minima, which has been shown to yield improved generalization compared to standard optimizers such as AdamW.

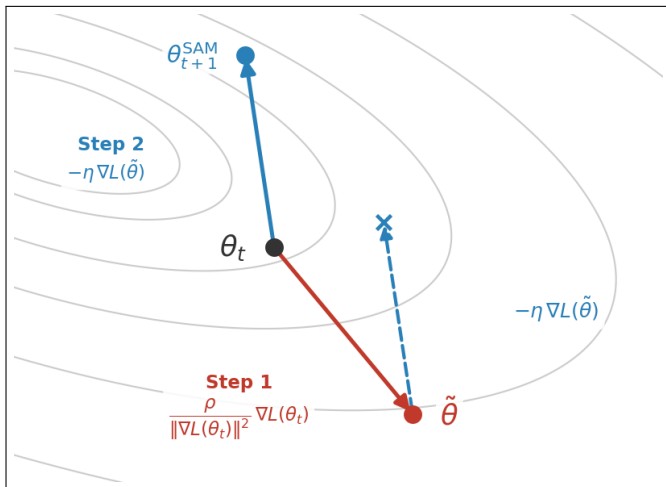

*Figure 10.* **SAM Update Schematic.** SAM first takes an ascent step along the gradient, evaluates the gradient at this perturbed point, and then updates the parameters using this perturbed gradient.

## A.2. Learning rate schedules

### A.2.1. COSINE

The cosine schedule (Loshchilov & Hutter, 2017) gradually anneals the learning rate following a cosine curve, allowing for large updates early in training and smaller, more stable updates near convergence. It is commonly combined with a warmup phase, during which the learning rate increases linearly from zero to the peak learning rate $\alpha_{\max}$ over $T_{\text{warmup}}$ steps. After warmup, the learning rate follows the cosine decay:

$$\alpha_t = \alpha_{\min} + \frac{1}{2}(\alpha_{\max} - \alpha_{\min})\left(1 + \cos\left(\frac{\pi(t - T_{\text{warmup}})}{T - T_{\text{warmup}}}\right)\right),$$

where $\alpha_{\min}$ is the minimum learning rate, often set to zero, $T$ is the total number of training steps, and $t > T_{\text{warmup}}$.

### A.2.2. WARMUP-STABLE-DECAY

This warmup-stable-decay (WSD) schedule (Hu et al., 2024) consists of three phases: an initial warmup phase where the learning rate increases linearly from zero to a peak value, a stable phase where the learning rate is held constant, and a final decay/anneal phase where the learning rate is gradually reduced. Warmup mitigates optimization instability caused by large gradients at initialization, the stable phase enables effective learning at a fixed scale, and the decay phase promotes convergence.

## B. Experimental details: OLMo-2-1B experiments

### B.1. Mid-training

We take an OLMo-2-1B (OLMo et al., 2024) checkpoint[1] which was pretrained on 4T tokens and then mid-train it for 50B tokens on the Dolmino mixture (OLMo et al., 2024) using AdamW and SAM. We select $\rho = 0.05$ for SAM (defined in Section 2.3.1), which we determined by tuning preliminary small-scale experiments.

### B.1.1. TRAINING CONFIGURATION

We use the same model architecture and training configuration as defined in OLMo et al. (2024), with the small change of reducing the maximum context length from 4096 to 2048 and accordingly doubling the batch size to keep the total tokens seen in each step the same. This was done to fit our GPU constraints. The final configuration is given in Table 1.

---

[1]https://github.com/allenai/OLMo/blob/main/configs/official-0425/OLMo-2-0425-1B.csv

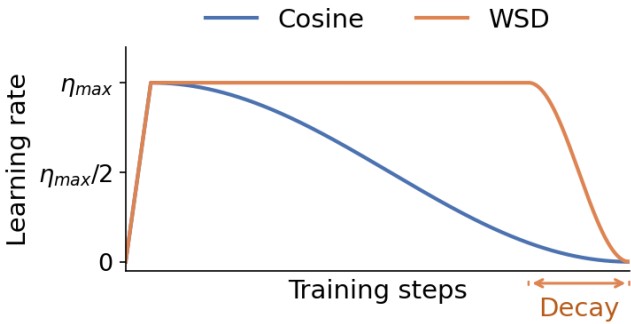

*Figure 11.* **Learning rate scheduling schematic for Cosine and Warmup-Stable-Decay (WSD)**

*Table 1.* Mid-training configuration used in our OLMo-2-1B experiments.

| Configuration | OLMo-2-1B |
|---|---|
| Seed | 42 |
| Layers | 16 |
| Heads | 16 |
| Vocab Size | 100278 |
| Embedding Size | 100352 |
| Hidden dimensions | 2048 |
| Max context length | 2048 |
| Activation type | SwiGLU |
| Attention dropout | 0.0 |
| Residual dropout | 0.0 |
| Embedding dropout | 0.0 |
| Beta1 | 0.9 |
| Beta2 | 0.95 |
| Warmup steps | 0 |
| Weight decay | 0.1 |
| Batch size | 1024 |
| Learning rate | 7.4487e-5 |

### B.1.2. EVALUATION

We use the `OLMES` framework[2] for evaluation on the OLMo pretraining benchmark suite: ARC-Challenge (Clark et al., 2018), HellaSwag (Zellers et al., 2019), MMLU (Hendrycks et al., 2020), Winogrande (Sakaguchi et al., 2019), DROP (Dua et al., 2019), Natural Questions (Kwiatkowski et al., 2019), AGIEval-English (Zhong et al., 2023), GSM8K (Cobbe et al., 2021), MMLU-Pro (Wang et al., 2024b), and TriviaQA (Joshi et al., 2017). We report the model performance after mid-training in Table 2 for AdamW (*OLMo baseline*) and SAM. We can see that our versions of the mid-trained models roughly match the numbers reported by OLMo et al. (2024) in Table 9.

*Table 2.* Mid-training OLMo benchmark results for OLMo-2-1B

| OLMo-2-1B | Avg | MMLU | $ARC_C$ | HSwag | WinoG | NQ | DROP | AGIEval | GSM8K | $MMLU_{PRO}$ | TQA |
|---|---|---|---|---|---|---|---|---|---|---|---|
| OLMo | 43.7 | 44.3 | 51.3 | 69.5 | 66.5 | 20.8 | 34.0 | 36.3 | 43.8 | 16.1 | 54.7 |
| AdamW | 43.2 | 44.3 | 47.4 | 69.4 | 67.4 | 20.90 | 33.40 | 34.8 | 44.0 | 15.4 | 55.1 |
| SAM | 42.9 | 44.4 | 47.1 | 69.2 | 67.3 | 20.4 | 32.9 | 35.7 | 42.5 | 15.5 | 54.40 |

---

[2]https://github.com/allenai/olmes

## B.2. Fine-tuning

We fine-tune the different mid-trained checkpoints on four publicly available datasets: MetaMath (Yu et al., 2023) and StackMathQA (Zhang, 2024) (mathematical reasoning), Tülu-3 (Lambert et al., 2025) (instruction following), and MusicPile (Yuan et al., 2024) (domain-specific). We use the AdamW optimizer with a cosine learning rate schedule. To estimate the learning–forgetting tradeoff set, we sweep over learning rates ranging from $2e-6$ to $2e-4$. We fine-tune for 1 epoch on MetaMath (80M) and for 50M tokens on the other three datasets. The detailed hyperparameters can be seen in Table 3.

*Table 3.* Fine-tuning hyperparameters used in our OLMo-2-1B experiments.

| Hyperparameters | Values |
| --- | --- |
| Learning rate | $2e-6, 3e-6, 4e-6, 5e-6, 6e-6, 7e-6, 8e-6, 1e-5, 2e-5, 3e-5, 4e-5, 5e-5,$ |
| | $6e-5, 7e-5, 8e-5, 9e-5, 1e-4, 1.25e-4, 1.5e-4, 2e-4$ |
| Batch size | 64 |
| Learning rate scheduler | Cosine |
| Optimizer | AdamW |
| Weight decay | 0.0 |
| Warmup steps | 10% of training |

## B.3. Evaluation

To evaluate forgetting, we compare the average benchmark (same as used in Appendix B.1.2) performance before and after post-training the mid-trained checkpoint. We chose the benchmark performance at a reasonable fine-tuning loss tradeoff (depicted by the horizontal line in Figure 13) to measure forgetting.

# C. Experimental details: controlled experiments

## C.1. Pretraining

We tune the learning rate for each model checkpoint individually to minimize the pretraining validation loss. For SAM, we select $\rho = 0.05$ (defined in Section 2.3.1), which we determined by tuning preliminary small-scale model checkpoints.

### C.1.1. TRAINING CONFIGURATION

We pretrain models from scratch at three parameter scales: 20M, 60M, and 150M using the OLMo architecture (Groeneveld et al., 2024) and pretraining recipe. The model configurations can be seen in Table 4. Each model is trained with varying token budgets (Table 5), corresponding to token to parameter ratios of 100 to 3200, on the DCLM web data (Li et al., 2024). We evaluate two optimizers, AdamW (Loshchilov & Hutter, 2019), and Sharpness-Aware Minimization (Foret et al., 2021) in combination with two learning rate schedules: cosine (Loshchilov & Hutter, 2017) and warmup-stable-decay (WSD) (Hu et al., 2024).

*Table 4.* Pretraining model configuration used in our controlled experiments.

| Configuration | OLMo-20M | OLMo-60M | OLMo-150M |
|---|---|---|---|
| Layers | 8 | 8 | 12 |
| Heads | 8 | 8 | 12 |
| Vocab Size | 100278 | 100278 | 100278 |
| Embedding Size | 100352 | 100352 | 100352 |
| Hidden dimensions | 256 | 528 | 768 |
| Max context length | 1024 | 1024 | 1024 |
| Activation type | SwiGLU | SwiGLU | SwiGLU |
| Attention dropout | 0.0 | 0.0 | 0.0 |
| Residual dropout | 0.0 | 0.0 | 0.0 |
| Embedding dropout | 0.0 | 0.0 | 0.0 |
| Beta1 | 0.9 | 0.9 | 0.9 |
| Beta2 | 0.95 | 0.95 | 0.95 |
| Warmup steps | 1000 | 2000 | 3000 |
| Weight decay | 0.1 | 0.1 | 0.1 |
| Batch size | 256 | 256 | 256 |

*Table 5.* Pretraining token budgets used in our controlled experiments.

| Model | Token Budgets (B) |
|---|---|
| OLMo-20M | $4, 8, 16, 32, 64$ |
| OLMo-60M | $12, 24, 48, 96, 192$ |
| OLMo-150M | $15, 30, 60, 120$ |

### C.1.2. LR TUNING

We tune the learning rate of the pretrained models with AdamW and Cosine schedule and use the best value found for SAM (which uses AdamW as the base optimizer) and WSD schedule. For each model size, we start with the smallest token budget and sweep over the learning rates $\in [1e-4, 3e-4, 6e-4, 1e-3, 3e-3, 1e-2]$ to find the one which has the lowest pretraining loss on a held out validation set. For the next token budget, we only look at smaller LRs to check if it's better, based on observations from past work which show that optimal learning rate decreases with increasing token budgets (Bjorck et al., 2025). Our final learning rates used for each combination of model size and tokens per parameter can be seen in Table 6 and the corresponding schematic of tuning can be seen in Figure 12.

*Table 6.* Final pretraining learning rates for different model sizes for varying tokens to parameter ratios. We use the same pretraining learning rate with SAM (which uses AdamW as the base optimizer) and with WSD schedule.

| Tokens / Param | 20M | 60M | 150M |
|---|---|---|---|
| 100 | – | – | 0.001 |
| 200 | 0.003 | 0.001 | 0.001 |
| 400 | 0.003 | 0.001 | 0.0006 |
| 800 | 0.003 | 0.0006 | 0.0003 |
| 1600 | 0.001 | 0.0006 | – |
| 3200 | 0.001 | 0.0003 | – |

### C.2. Fine-tuning

We fine-tune the different pretrained checkpoints on five publicly available datasets: StarCoder (Li et al., 2023) (code generation), GSM8K (Cobbe et al., 2021) and StackMathQA (Zhang, 2024) (mathematical reasoning), Tülu-3 (Lambert et al., 2025) (instruction following), and MusicPile (Yuan et al., 2024) (domain-specific). We use the AdamW optimizer with

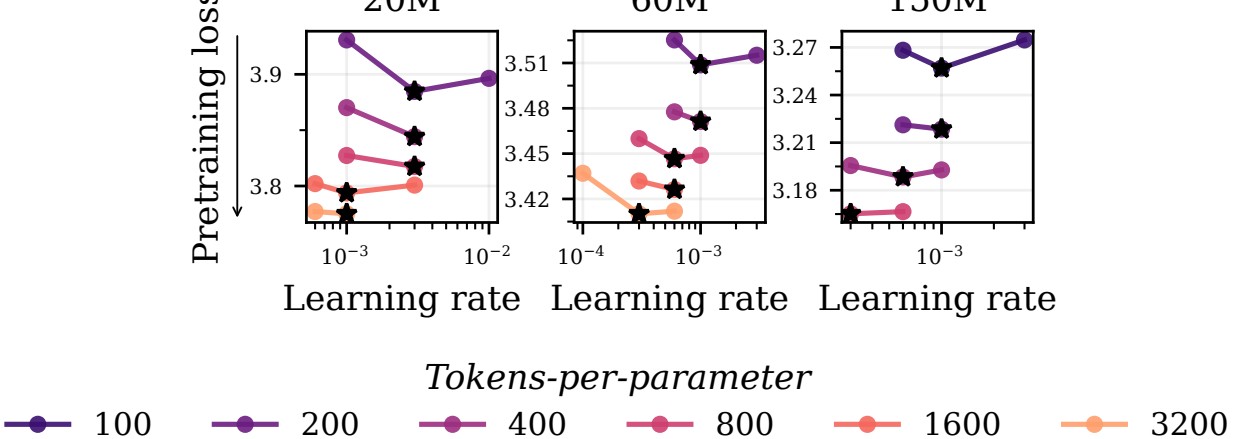

*Figure 12.* Pretraining learning rate tuning for controlled experiments.

a cosine learning rate schedule. To estimate the learning–forgetting tradeoff set, we sweep over learning rates ranging from $1e-6$ to $1e-2$. We tune batch size and weight decay to find 64 and 0 as the optimal values, respectively. We fine-tune on each dataset for one epoch or a maximum of 10M tokens. This results in 1 epoch for StackMathQA (1.2M) and 10M for all four other datasets. The detailed hyperparameters can be seen in Table 7.

*Table 7.* Fine-tuning hyperparameters used in our controlled experiments.

| Hyperparameters | Values |
|---|---|
| Learning rate | $1e-6, 2e-6, 4e-6, 8e-6, 1e-5, 2e-5, 3e-5, 4e-5, 5e-5, 6e-5, 7e-5, 8e-5,$ |
| | $9e-5, 1e-4, 1.1e-4, 1.2e-4, 1.25e-4, 1.4e-4, 1.5e-4, 1.6e-4, 1.8e-4, 2e-4,$ |
| | $2.4e-4, 2.5e-4, 3e-4, 3.5e-4, 4e-4, 5e-4, 6e-4, 7e-4, 8e-4, 9e-4, 1e-3,$ |
| | $1.25e-3, 1.5e-3, 2e-3, 2.5e-3, 3e-3, 4e-3, 5e-3, 6e-3, 8e-3, 1e-2$ |
| Batch size | 32, 64*, 128 |
| Learning rate scheduler | Cosine |
| Optimizer | AdamW |
| Weight decay | 0.0*, 0.1 |
| Warmup steps | 10% of training |
| Tokens | Min(tokens in dataset, 10M) |

## C.3. Gaussian perturbations

To capture *average-case* degradation over perturbation directions, we apply isotropic Gaussian noise directly to the pretrained weights. For each layer $\ell$ with weight tensor $W^{(\ell)}$, we sample $Z^{(\ell)} \sim \mathcal{N}(0, I)$ and form the perturbed weights

$$\tilde{W}^{(\ell)} \;=\; W^{(\ell)} \;+\; \gamma \cdot \|W^{(\ell)}\|_F \cdot \frac{Z^{(\ell)}}{\|Z^{(\ell)}\|_F}, \tag{6}$$

where $\gamma \in \{0.009, \ 0.013, \ 0.017, \ 0.020, \ 0.025\}$ controls the perturbation magnitude.

## C.4. Evaluation at a fixed fine-tuning loss

For a given pretrained checkpoint $\theta_{\mathrm{PT}}$, we define the learning–forgetting tradeoff set as

$$\mathcal{T}(\theta_{\mathrm{PT}}) = \left\{ \left( \mathcal{L}_{\mathrm{FT}}(\theta_{\mathrm{FT}}), \mathcal{L}_{\mathrm{PT}}(\theta_{\mathrm{FT}}) \right) \;\middle|\; \theta_{\mathrm{FT}} \in \Theta_{\mathrm{FT}}(\theta_{\mathrm{PT}}) \right\}. \tag{7}$$

To enable loss-matched comparison across pretrained checkpoints, we define a common fine-tuning loss threshold as follows. For each checkpoint $\theta_{\mathrm{PT}}^{(i)}$, we first compute the minimum fine-tuning loss achieved within its tradeoff set:

$$\mathcal{L}_{\min}^{(i)} = \min_{(\mathcal{L}_{\mathrm{FT}}, \mathcal{L}_{\mathrm{PT}}) \in \mathcal{T}(\theta_{\mathrm{PT}}^{(i)})} \mathcal{L}_{\mathrm{FT}}. \tag{8}$$

We then define the global fine-tuning threshold $\tau$ as the maximum over these per-checkpoint minima:

$$\tau = \max_i \; \mathcal{L}_{\min}^{(i)}. \tag{9}$$

For each pretrained checkpoint, we report the retained pretraining loss $\mathcal{L}_{\mathrm{PT}}$ corresponding to the model on its tradeoff frontier whose fine-tuning loss satisfies $\mathcal{L}_{\mathrm{FT}} \leq \tau$.

# D. Additional results for OLMo-2-1B experiments

## D.1. Post-training quantization

*Table 8.* Task-wise scores on OLMo benchmark suite for OLMo-2-1B mid-trained on AdamW and SAM after 4-bit quantization.

| OLMo-2-1B | Avg | MMLU | ARC$_C$ | HSwag | WinoG | NQ | DROP | AGIEval | GSM8K | MMLU$_{PRO}$ | TQA |
|---|---|---|---|---|---|---|---|---|---|---|---|
| AdamW (Base) | 43.2 | 44.30 | 47.40 | 69.40 | 67.40 | 20.90 | 33.40 | 34.80 | 44.00 | 15.40 | 55.10 |
| SAM (Base) | 42.9 | 44.40 | 47.10 | 69.20 | 67.30 | 20.40 | 32.90 | 35.70 | 42.50 | 15.50 | 54.40 |
| AdamW (4-bit) | 38.89 | 42.20 | 45.60 | 67.00 | 65.10 | 17.80 | **31.40** | 33.80 | 21.60 | 14.20 | 50.20 |
| SAM (4-bit) | **40.60** | **42.60** | **46.80** | **67.10** | **66.90** | **18.20** | 30.20 | 33.80 | **35.20** | 14.20 | **51.00** |

## D.2. Learning-forgetting frontier

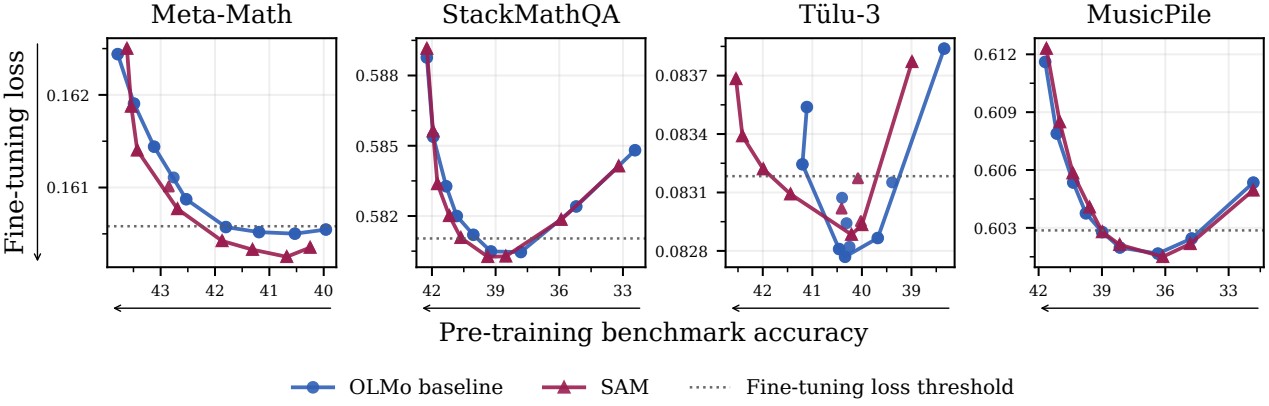

*Figure 13.* **Learning-forgetting frontier for OLMo-2-1B across MetaMath, StackMathQA, Tülu-3, and MusicPile.**

# E. Additional results for controlled experiments

*Table 9.* Summary of additional results for controlled experiments

| Optimization choice | Experiment | OLMo-20M | OLMo-60M | OLMo-150M |
|---|---|---|---|---|
| **Optimizer** | LF Frontier across Datasets | Figure 14 | Figure 2 | Figure 15 |
| | LF Frontier with Pretraining Tokens | Figures 16-20 | Figures 21-25 | Figures 26-30 |
| | LF Tradeoff at Matched Pretrain Loss | Figure 36 | Figure 5 | Figure 37 |
| | LF Frontier with EWC | - | Figures 38 & 39 | - |
| | Post-training Quantization | Figure 40 | Figure 41 | Figure 42 |
| | Gaussian Perturbation | Figure 43 | Figure 44 | Figure 45 |
| **Peak LR** | Base-model pretraining loss (WSD) | - | Figure 46 | - |
| | LF Frontier across Datasets | - | Figures 47 & 48 | - |
| | Gaussian Perturbation | - | Figure 49 | - |
| | Post-training Quantization | - | Figures 50 & 51 | - |
| **Anneal Percent** | LF Frontier across Datasets | - | Figure 52 | - |
| **Annealing with SAM** | LF Frontier across Datasets | Figure 53 | Figure 54 | Figure 55 |
| | Gaussian Perturbation | Figure 56 | Figure 57 | Figure 58 |
| | Post-training Quantization | Figure 59 | Figure 60 | Figure 61 |
| | | | **TPP-800** | |
| **Optimizer** | LF Frontier across Model Size | | Figures 31-34 | |

## E.1. Optimization choice: optimizer

### E.1.1. LEARNING-FORGETTING FRONTIER ACROSS DATASETS

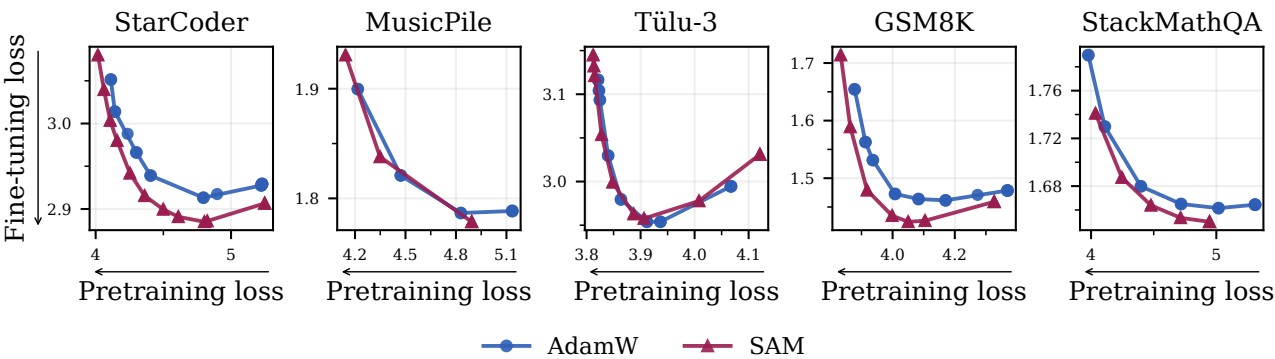

*Figure 14.* **AdamW vs. SAM learning-forgetting frontier for OLMo-20M across datasets at 64B tokens**

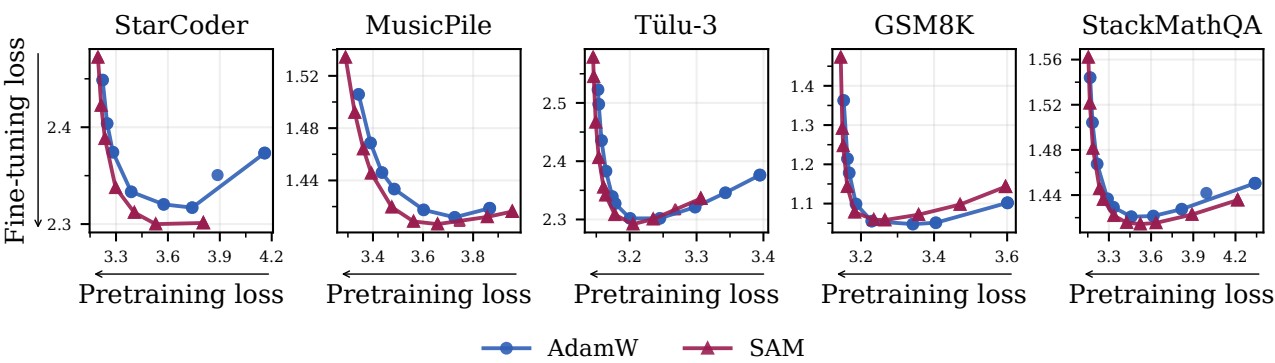

*Figure 15.* **AdamW vs. SAM learning-forgetting frontier for OLMo-150M across datasets at 240B tokens**

E.1.2. LEARNING-FORGETTING FRONTIER WITH SCALING PRETRAINING TOKENS

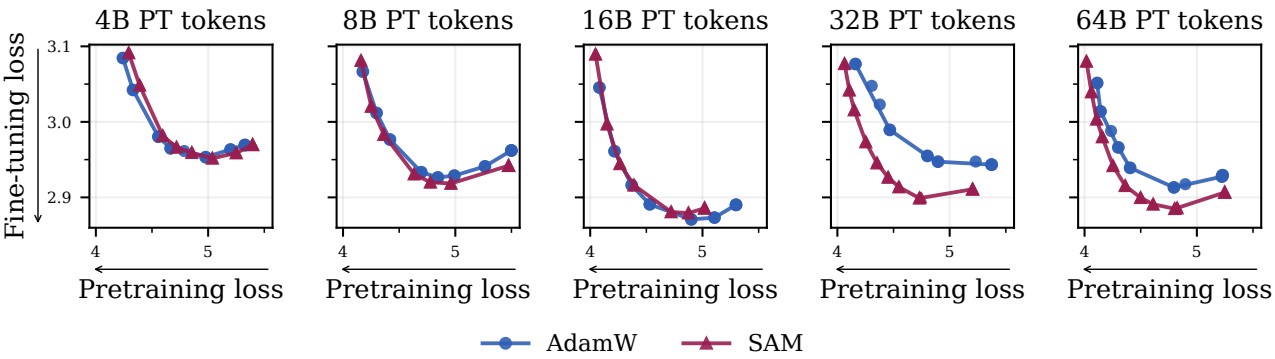

*Figure 16.* **AdamW vs. SAM with scaling pretraining tokens on StarCoder for OLMo-20M**

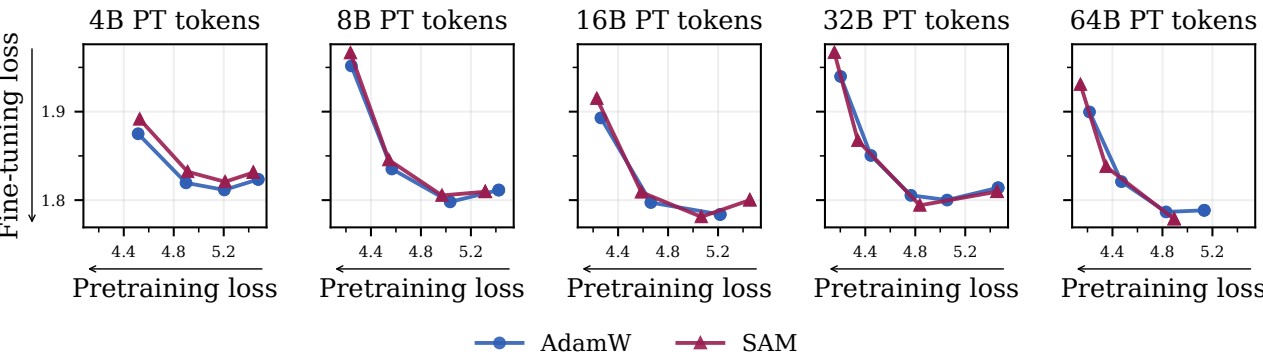

*Figure 17.* **AdamW vs. SAM with scaling pretraining tokens on MusicPile for OLMo-20M**

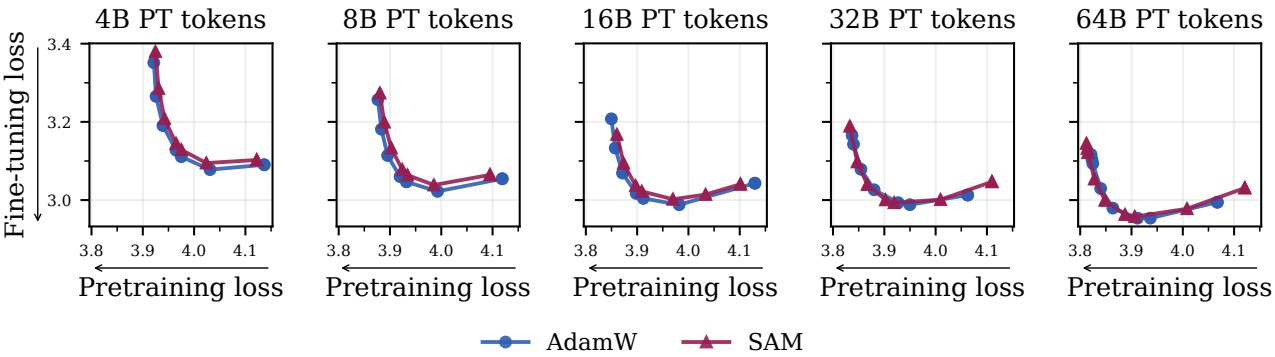

*Figure 18.* **AdamW vs. SAM with scaling pretraining tokens on Tülu-3 for OLMo-20M**

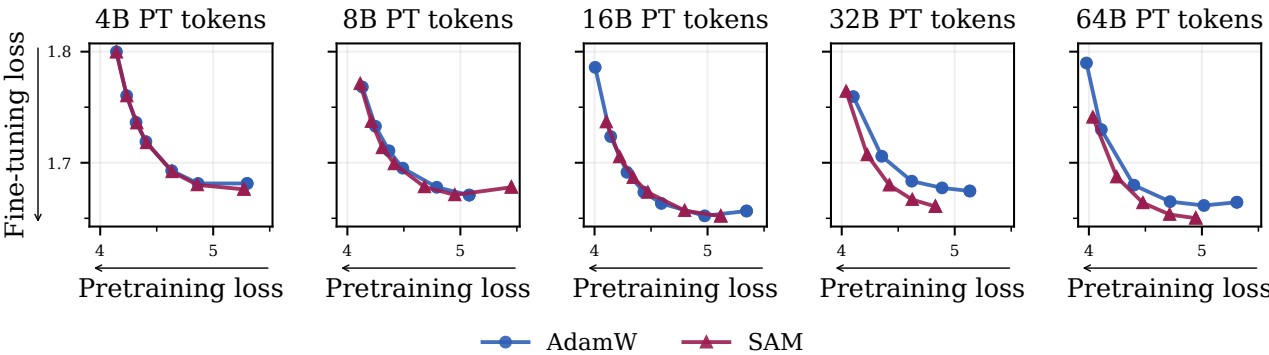

*Figure 19.* **AdamW vs. SAM with scaling pretraining tokens on StackMathQA for OLMo-20M**

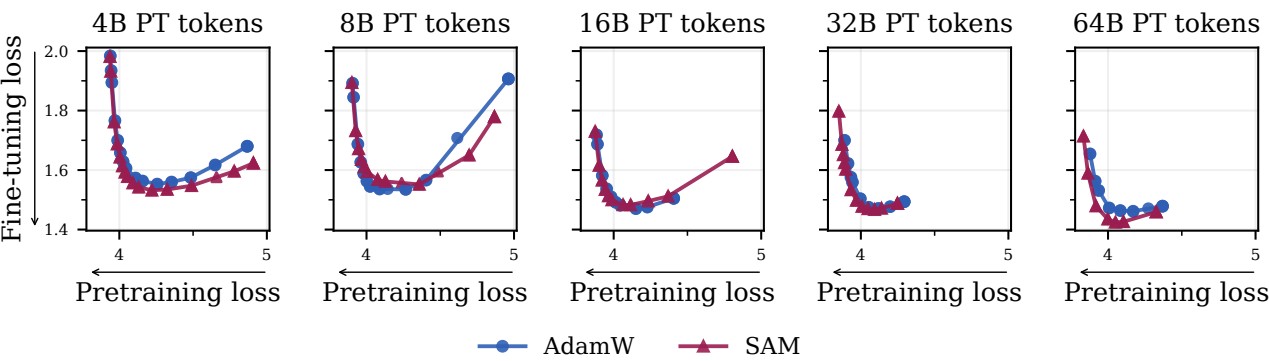

Figure 20. **AdamW vs. SAM with scaling pretraining tokens on GSM8K for OLMo-20M**

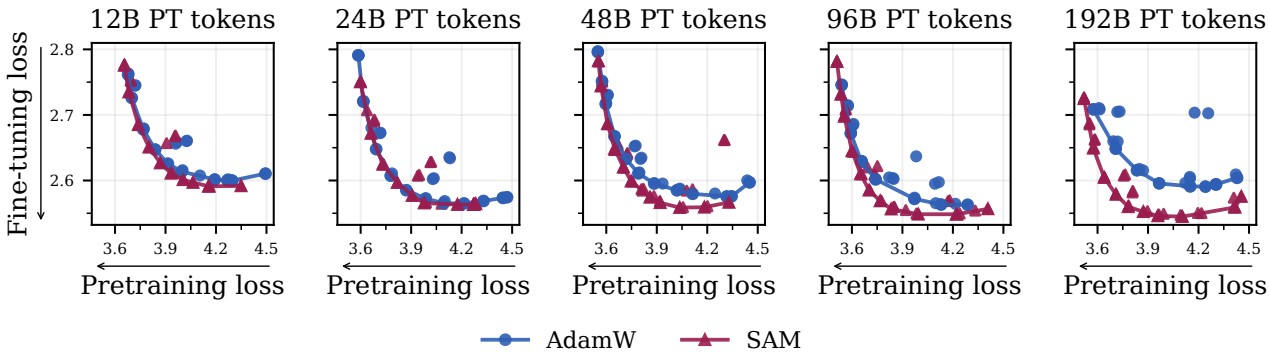

Figure 21. **AdamW vs. SAM with scaling pretraining tokens on StarCoder for OLMo-60M**

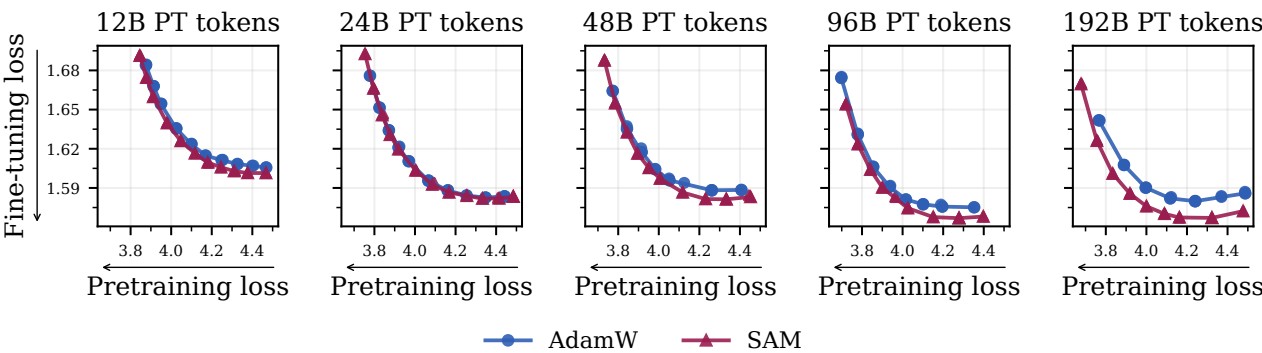

Figure 22. **AdamW vs. SAM with scaling pretraining tokens on MusicPile for OLMo-60M**

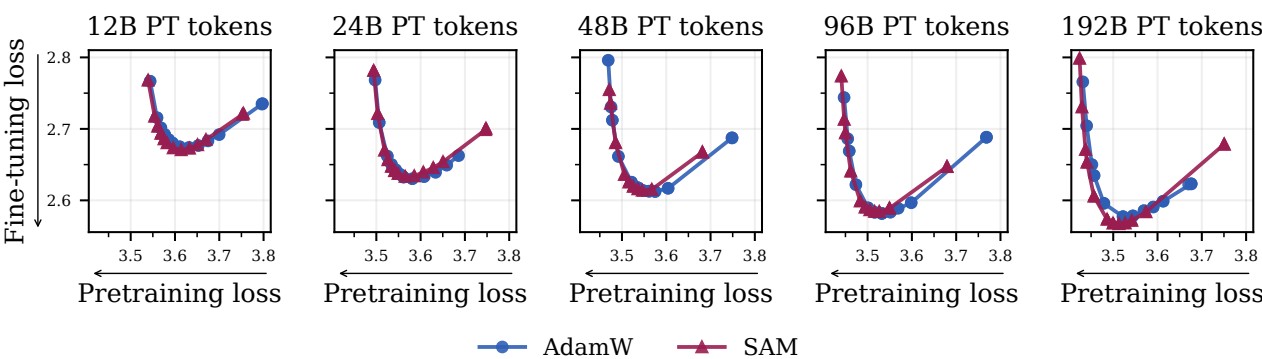

*Figure 23.* **AdamW vs. SAM with scaling pretraining tokens on Tülu-3 for OLMo-60M**

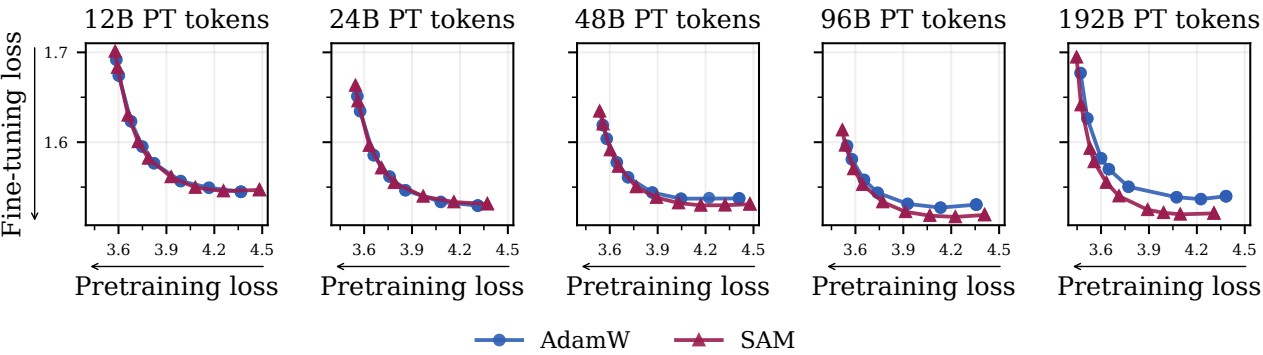

*Figure 24.* **AdamW vs. SAM with scaling pretraining tokens on StackMathQA for OLMo-60M**

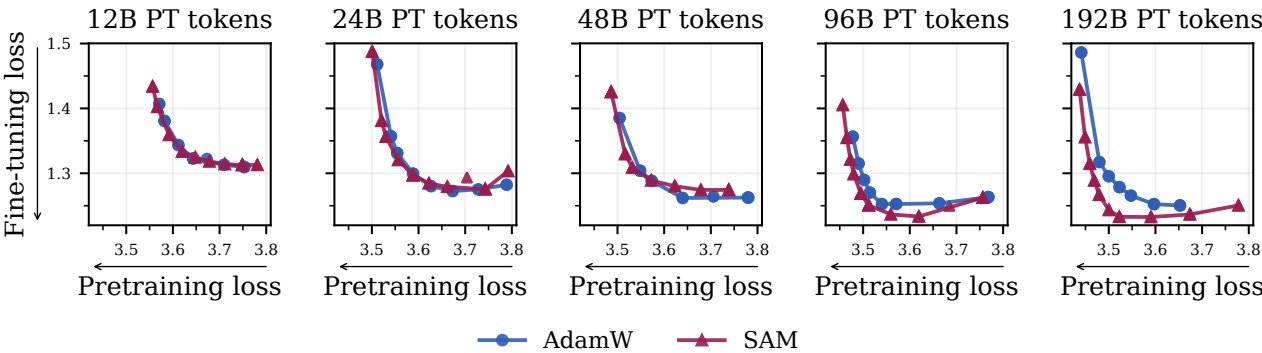

*Figure 25.* **AdamW vs. SAM with scaling pretraining tokens on GSM8K for OLMo-60M**

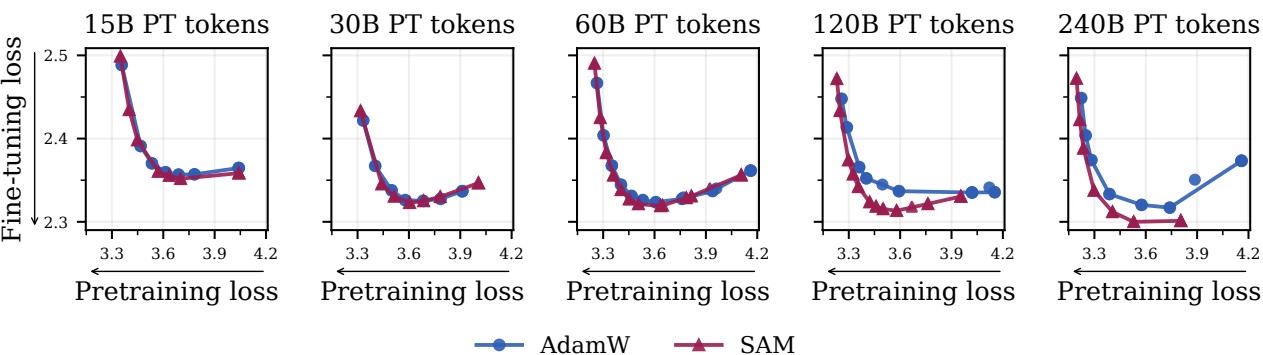

*Figure 26.* **AdamW vs. SAM with scaling pretraining tokens on StarCoder for OLMo-150M**

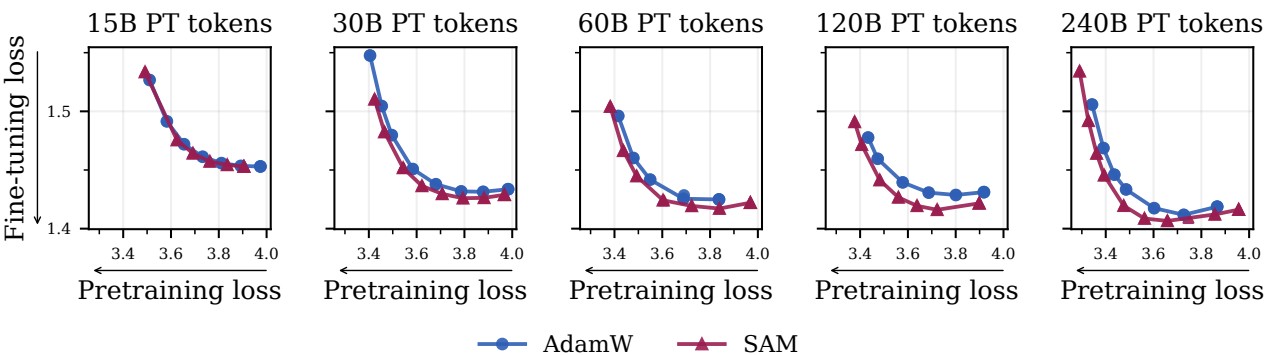

*Figure 27.* **AdamW vs. SAM with scaling pretraining tokens on MusicPile for OLMo-150M**

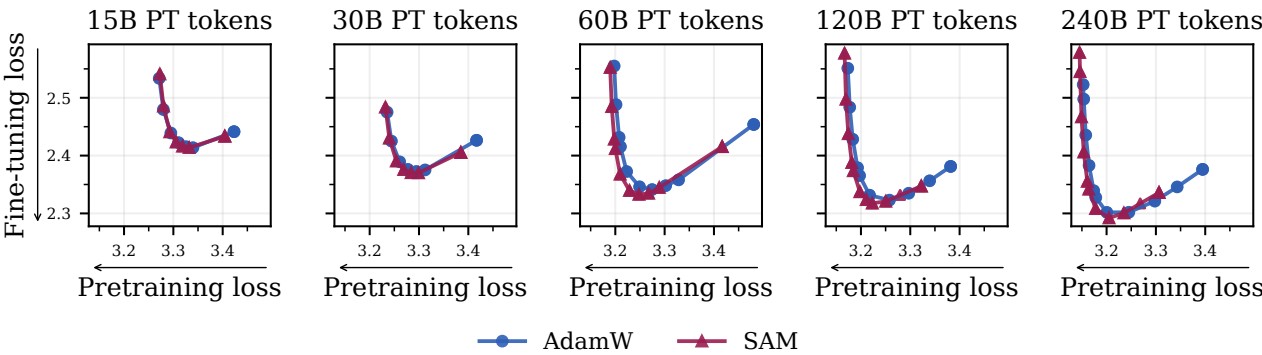

*Figure 28.* **AdamW vs. SAM with scaling pretraining tokens on Tülu-3 for OLMo-150M**

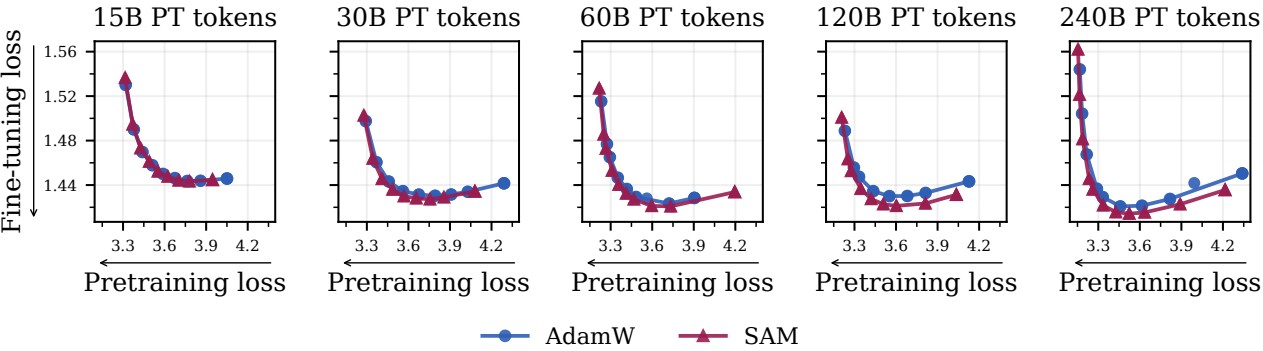

*Figure 29.* **AdamW vs. SAM with scaling pretraining tokens on StackMathQA for OLMo-150M**

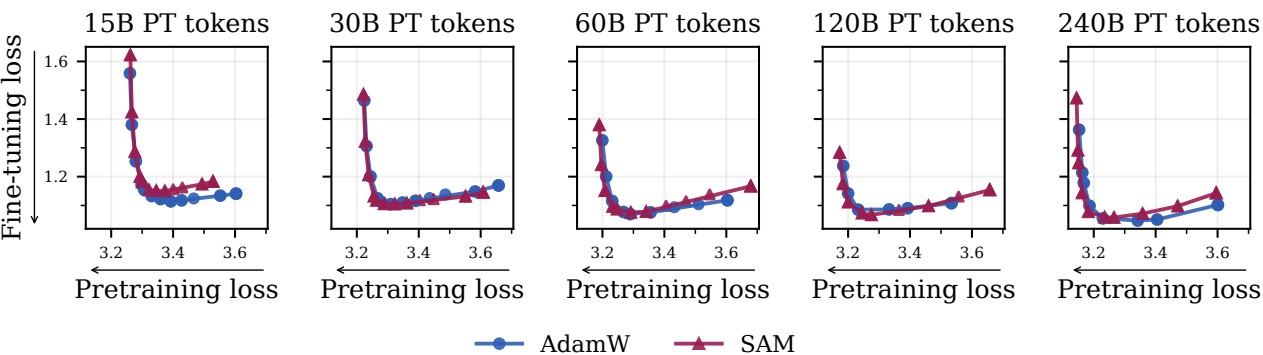

*Figure 30.* **AdamW vs. SAM with scaling pretraining tokens on GSM8K for OLMo-150M**

E.1.3. LEARNING-FORGETTING FRONTIER WITH MODEL SIZE

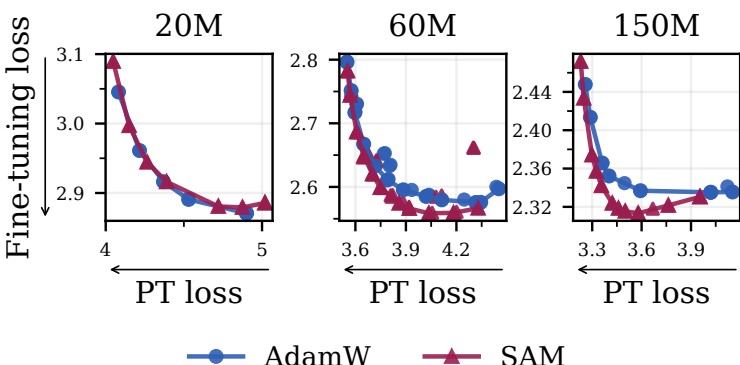

*Figure 31.* **AdamW vs. SAM learning-forgetting frontier across model sizes at 800** *token-per-parameter* **for StarCoder**

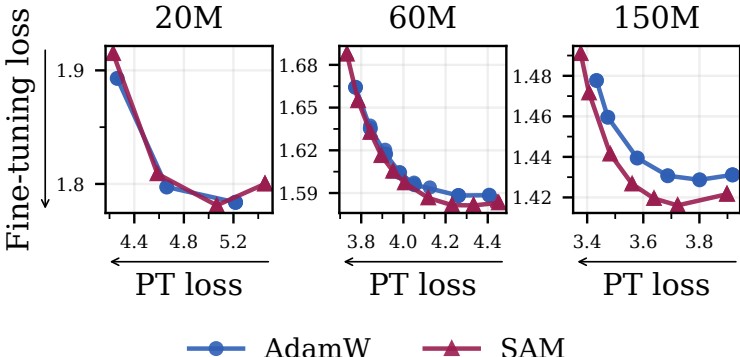

*Figure 32.* **AdamW vs. SAM learning-forgetting frontier across model sizes at 800** *token-per-parameter* **for MusicPile**

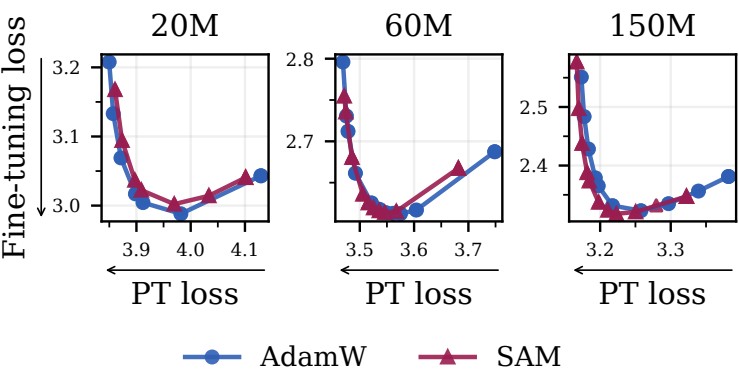

Figure 33. **AdamW vs. SAM learning-forgetting frontier across model sizes at 800** *token-per-parameter* **for Tülu-3**

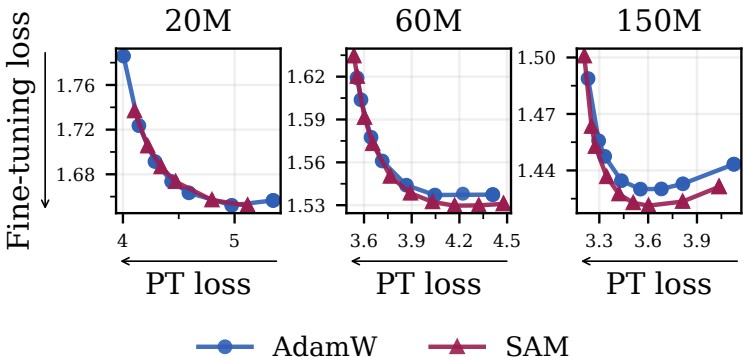

Figure 34. **AdamW vs. SAM learning-forgetting frontier across model sizes at 800** *token-per-parameter* **for StackMathQA**

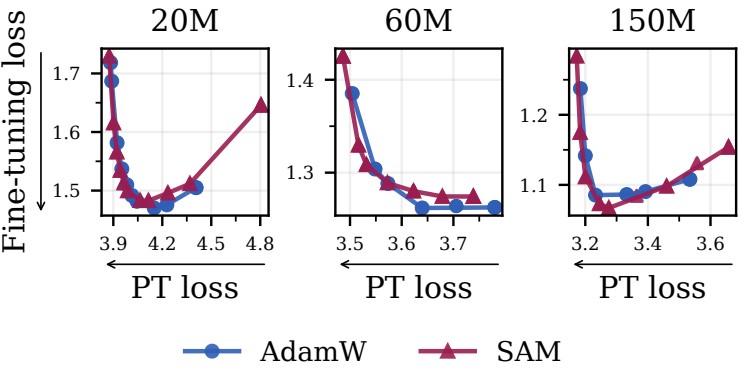

Figure 35. **AdamW vs. SAM learning-forgetting frontier across model sizes at 800** *token-per-parameter* **for GSM8K**

### E.1.4. LEARNING-FORGETTING TRADEOFF WITH MATCHED PRETRAINING LOSS

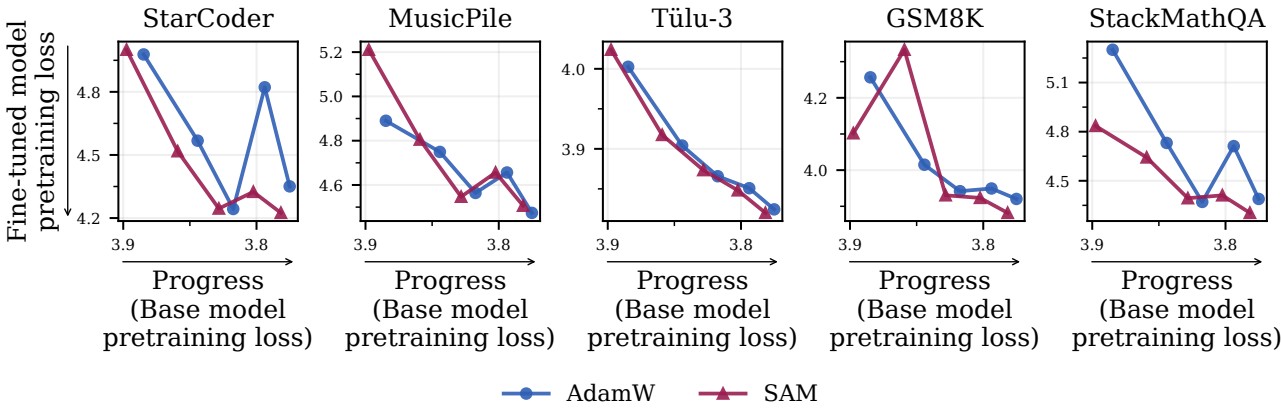

*Figure 36.* **Learning-forgetting tradeoff for AdamW vs. SAM at pretraining loss-matched setting for OLMo-20M across datasets.**

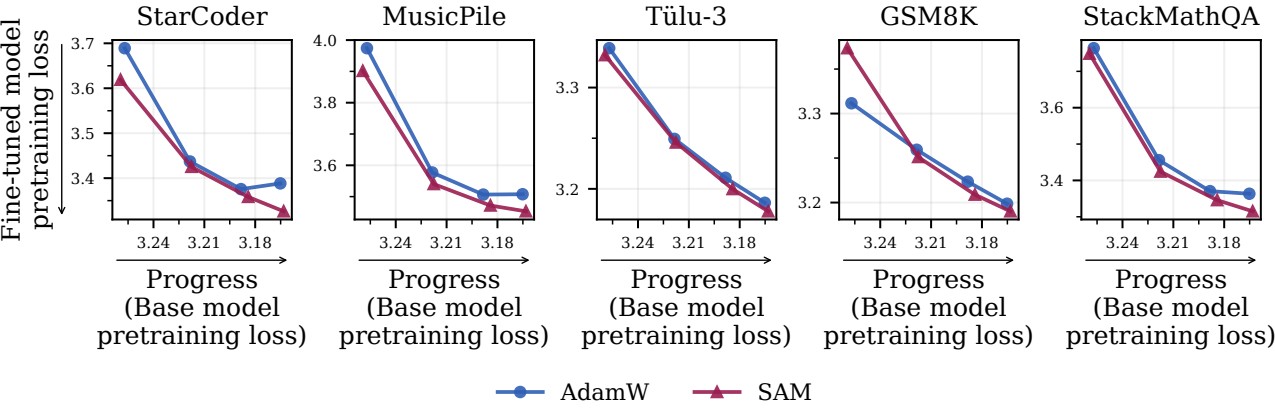

*Figure 37.* **Learning-forgetting tradeoff for AdamW vs. SAM at pretraining loss-matched setting for OLMo-150M across datasets.**

### E.1.5. LEARNING-FORGETTING TRADEOFF WITH EWC

To understand the effect of SAM in combination with other continual learning techniques, we use Elastic Weight Consolidation (EWC) (Kirkpatrick et al., 2017) for fine-tuning OLMo-60M checkpoints pretrained on 192B tokens with SAM.

**Tuning hyperparameters for EWC.** The EWC objective augments the current task loss $\mathcal{L}_{\text{new}}(\theta)$ with a quadratic penalty that keeps parameters $\theta$ close to their previous values $\theta^*$, weighted by their importance $F_i$ (estimated via the Fisher Information Matrix):

$$\mathcal{L}(\theta) = \mathcal{L}_{\text{new}}(\theta) + \lambda \sum_i F_i(\theta_i - \theta_i^*)^2 \tag{10}$$

Here, $\lambda$ controls the strength of this regularization. We tune $\lambda$ on StarCoder for both AdamW and SAM (Figure 38) and find $\lambda = 1e+4$ to be optimal. We then plot the learning–forgetting Pareto frontier using Table 10.

**SAM + EWC outperforms AdamW + EWC.** Combining SAM with EWC achieves a consistently better learning–forgetting Pareto frontier than AdamW with EWC, indicating that SAM's benefits persist under continual learning. This aligns with Mehta et al. (2023), which also shows that SAM benefits other continual learning techniques.

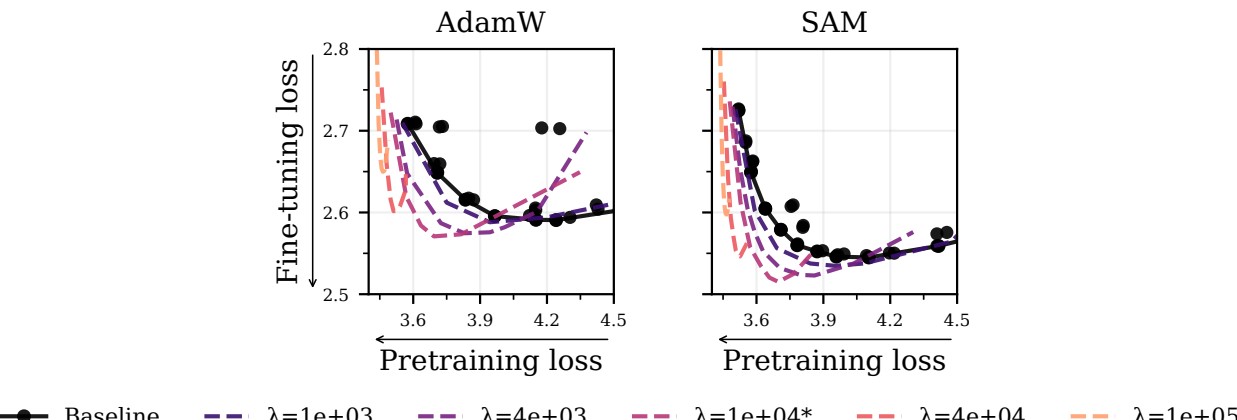

*Figure 38.* **Tuning $\lambda$ for EWC for OLMo-60M pretrained on 192B tokens and fine-tuned on StarCoder.** We pretrain OLMo-60M on 192B tokens using SAM and AdamW, and then fine-tune with Elastic Weight Consolidation (EWC) on StarCoder to tune $\lambda$ separately for each optimizer. We find $\lambda = 1e+04$ yields the best learning-forgetting Pareto frontier.

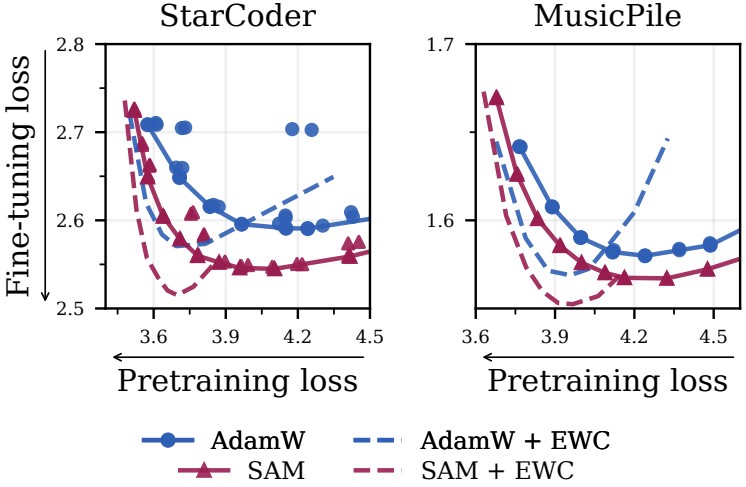

*Figure 39.* **SAM + EWC outperforms AdamW + EWC.** Learning-forgetting frontier for OLMo-60M pretrained on 192B tokens and fine-tuned on StarCoder and MusicPile with EWC.

*Table 10.* Hyperparameters used for generating EWC learning-forgetting Pareto frontier.

| Hyperparameter | Values |
|---|---|
| Learning Rates | $1e{-}4, 1.5e{-}4, 2e{-}4, 2.5e{-}4, 3e{-}4, 3.5e{-}4, 4e{-}4, 5e{-}4, 6e{-}4, 7e{-}4, 8e{-}4, 1e{-}3$ |
| Loss Regularization ($\lambda$) | $1e{+}3, 4e{+}3, 1e{+}4^{*}, 4e{+}4, 1e{+}5$ |
| Fisher Samples | 100 batches |
| Batch Size | 64 |
| Sequence Length | 1024 |
| Calibration Tokens | $\sim 6.5\text{M}$ |

### E.1.6. POST-TRAINING QUANTIZATION PERFORMANCE

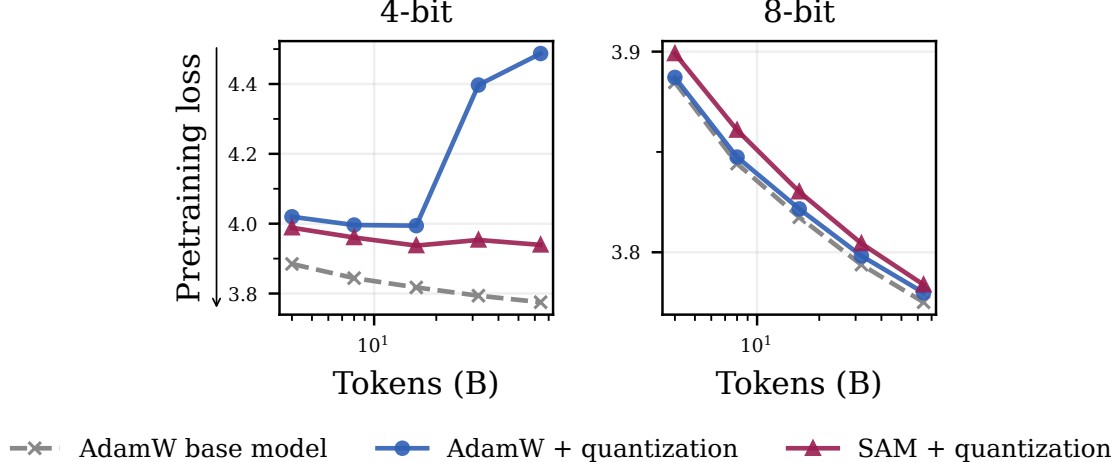

*Figure 40.* **AdamW vs. SAM under 4-bit and 8-bit post-training quantization for OLMo-20M.**

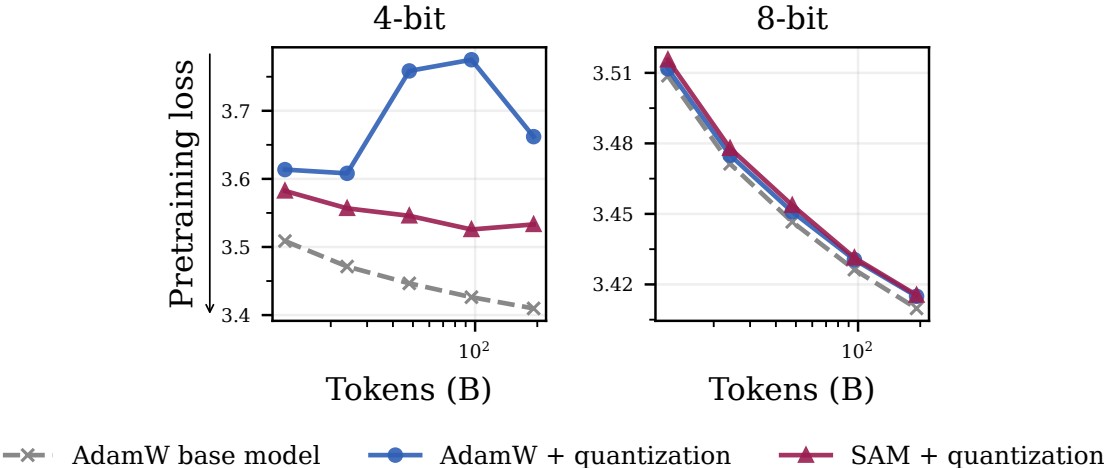

*Figure 41.* **AdamW vs. SAM under 4-bit and 8-bit post-training quantization for OLMo-60M.**

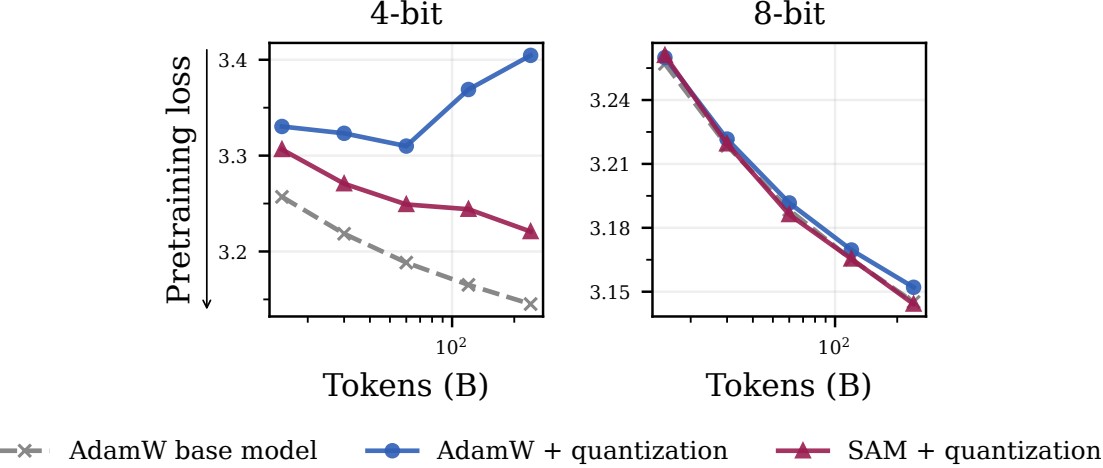

*Figure 42.* **AdamW vs. SAM under 4-bit and 8-bit post-training quantization for OLMo-150M.**

### E.1.7. GAUSSIAN PERTURBATION SENSITIVITY

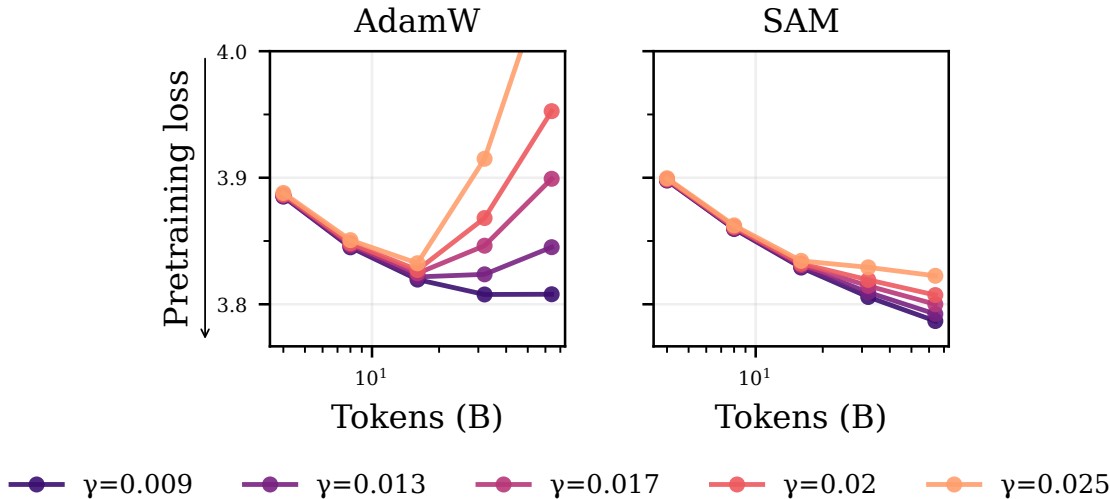

*Figure 43.* **AdamW vs. SAM Gaussian perturbation sensitivity for OLMo-20M.**

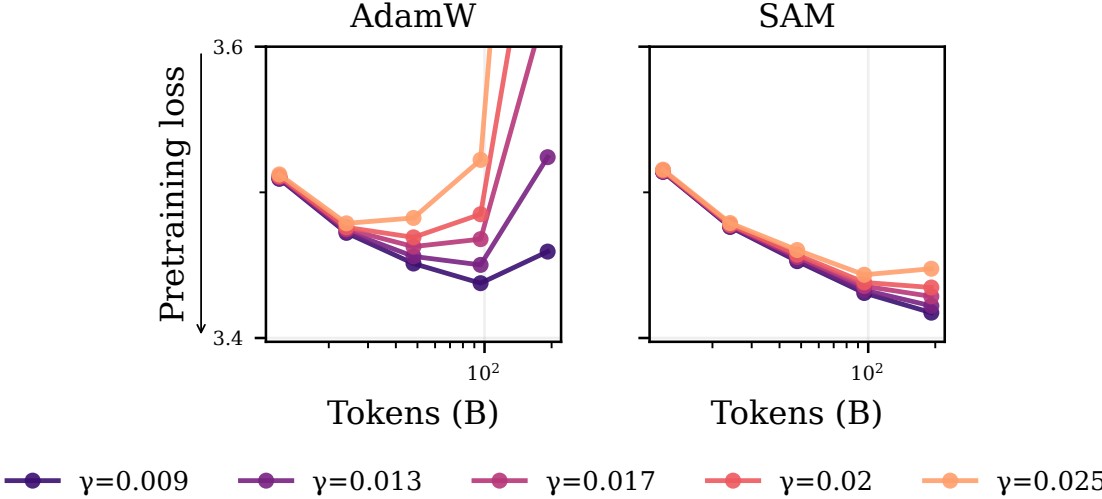

*Figure 44.* **AdamW vs. SAM Gaussian perturbation sensitivity for OLMo-60M.**

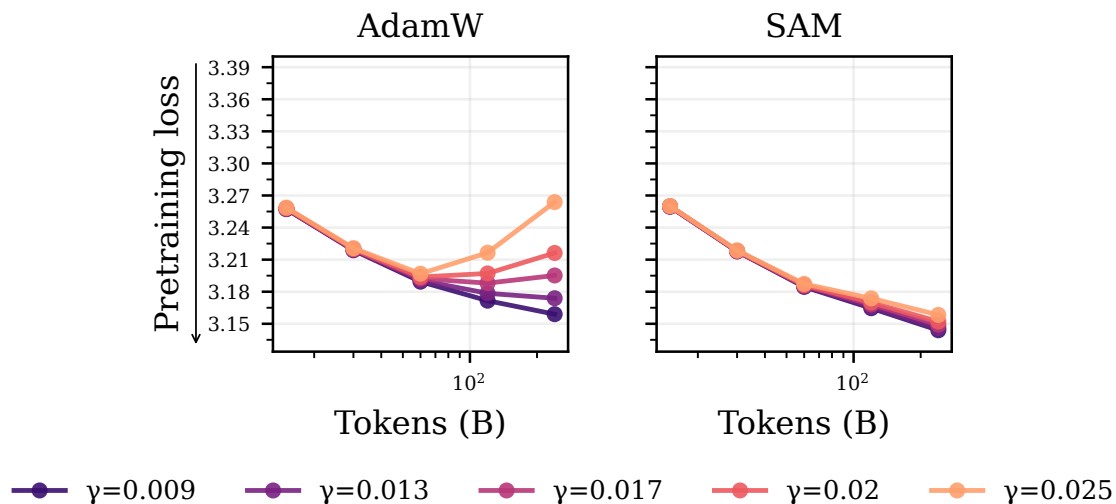

*Figure 45.* **AdamW vs. SAM Gaussian perturbation sensitivity for OLMo-150M.**

## E.2. Optimization choice: peak learning rate

We use OLMo-60M checkpoints pretrained on 192B tokens with different peak learning rates ($1e-4, 3e-4, 6e-4, 1e-3, 3e-3$) and learning rate schedules (cosine and WSD) as the base models for these experiments.

### E.2.1. BASE-MODEL PRETRAINING LOSS (WSD)

Base-model pretraining loss as a function of peak learning rate with WSD (10% annealing steps) for OLMo-60M pretrained on 192B tokens. The Cosine counterpart is the left panel of Figure 6 in the main paper.

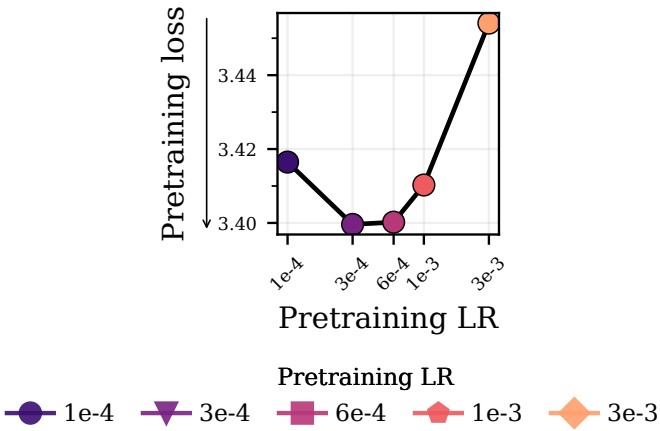

*Figure 46.* **Base-model pretraining loss across peak LR using WSD for OLMo-60M pretrained on 192B tokens.**

### E.2.2. LEARNING-FORGETTING FRONTIER ACROSS DATASETS

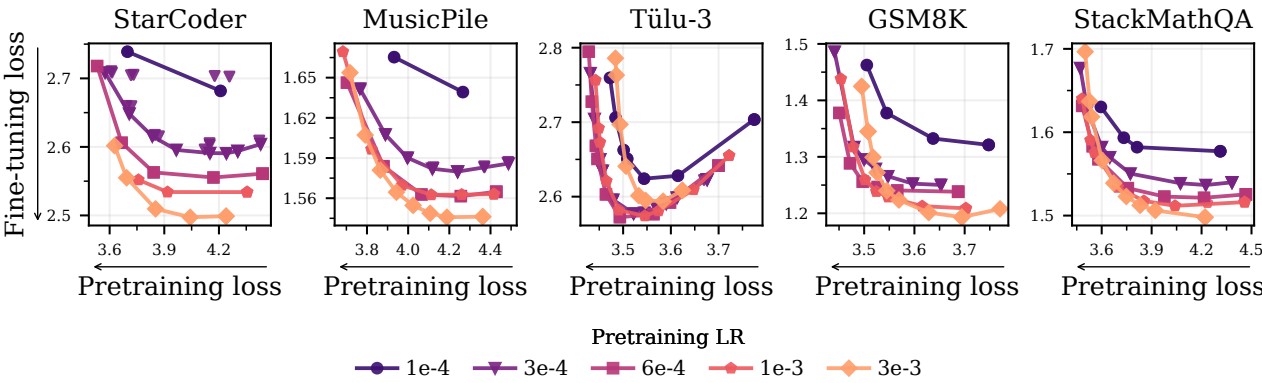

*Figure 47.* **Peak learning rate learning-forgetting frontier across datasets for cosine schedule.**

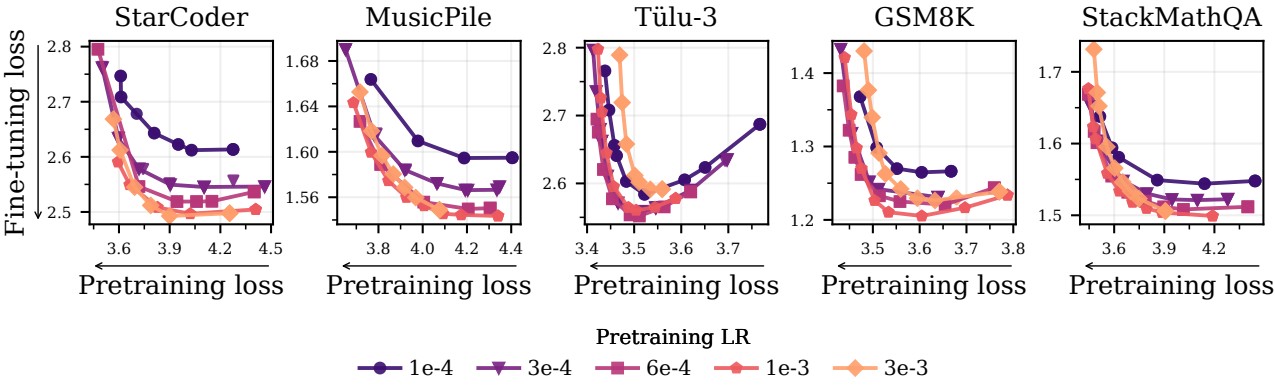

*Figure 48.* **Peak learning rate learning-forgetting frontier across datasets for WSD schedule (10% annealing steps).**

### E.2.3. GAUSSIAN PERTURBATION SENSITIVITY

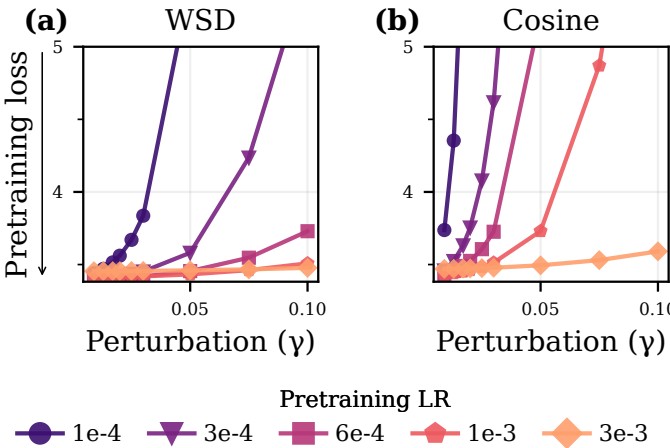

*Figure 49.* **Perturbed pretraining loss vs. perturbation magnitude $\gamma$ for a sweep of peak learning rates. (a) WSD (10% annealing steps) and (b) cosine schedule OLMo-60M pretrained on 192B tokens.**

### E.2.4. POST-TRAINING QUANTIZATION PERFORMANCE

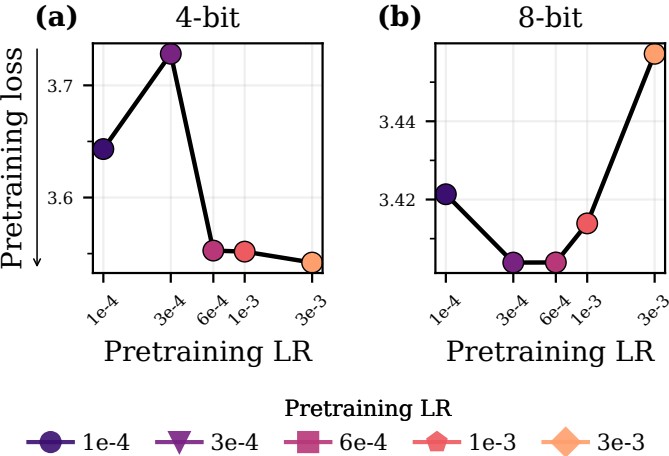

*Figure 50.* **Effect of peak learning rate with WSD schedule (10% annealing steps) for OLMo-60M pretrained on 192B tokens under 4-bit and 8-bit post-training quantization.**

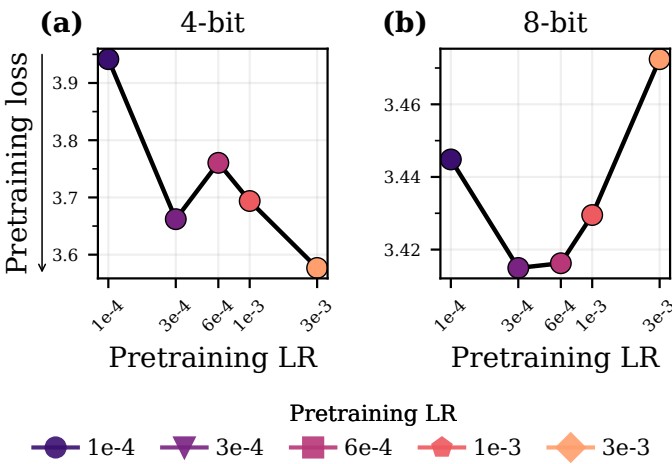

*Figure 51.* **Effect of peak learning rate with cosine schedule (10% annealing steps) for OLMo-60M pretrained on 192B tokens under 4-bit and 8-bit post-training quantization.**

## E.3. Optimization choice: annealing percent

### E.3.1. LEARNING-FORGETTING FRONTIER ACROSS DATASETS

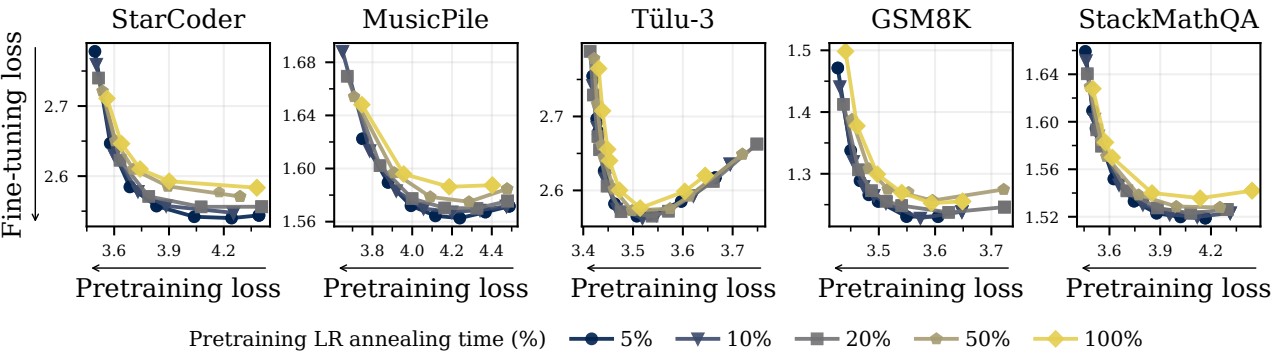

*Figure 52.* **Learning-forgetting frontier across datasets for OLMo-60M pretrained on 192B tokens using WSD schedule with varying periods of annealing.**

## E.4. Optimization choice: annealing with SAM

### E.4.1. LEARNING-FORGETTING FRONTIER ACROSS DATASETS

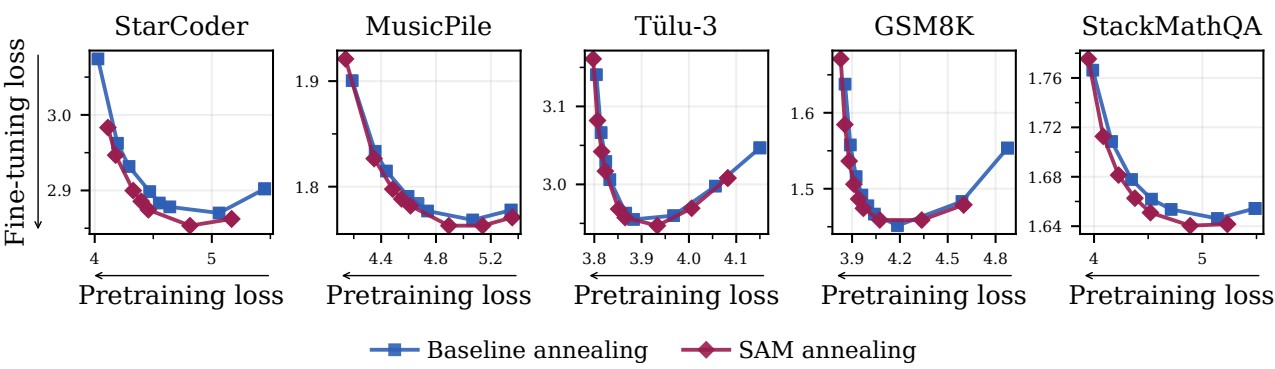

*Figure 53.* **Annealing with SAM vs. baseline annealing (WSD) learning-forgetting frontier across datasets for OLMo-20M pretrained on 64B tokens.**

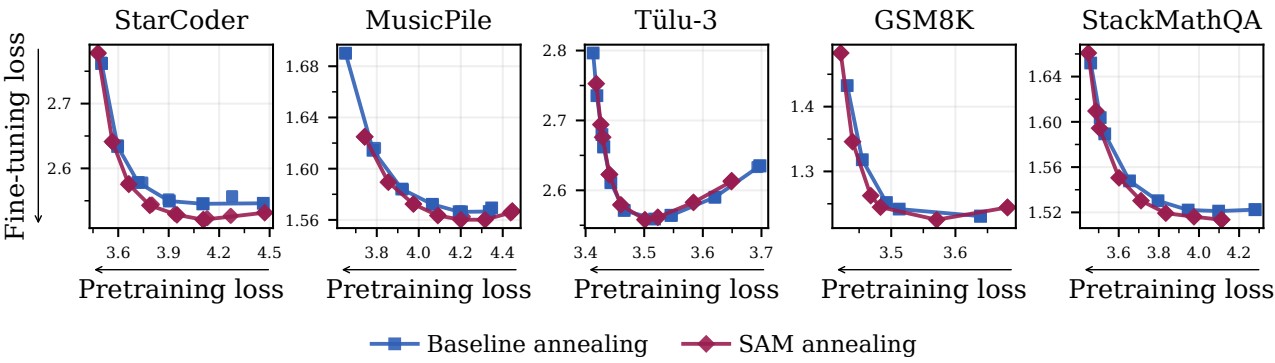

*Figure 54.* **Annealing with SAM vs. baseline annealing (WSD) learning-forgetting frontier across datasets for OLMo-60M pretrained on 192B tokens.**

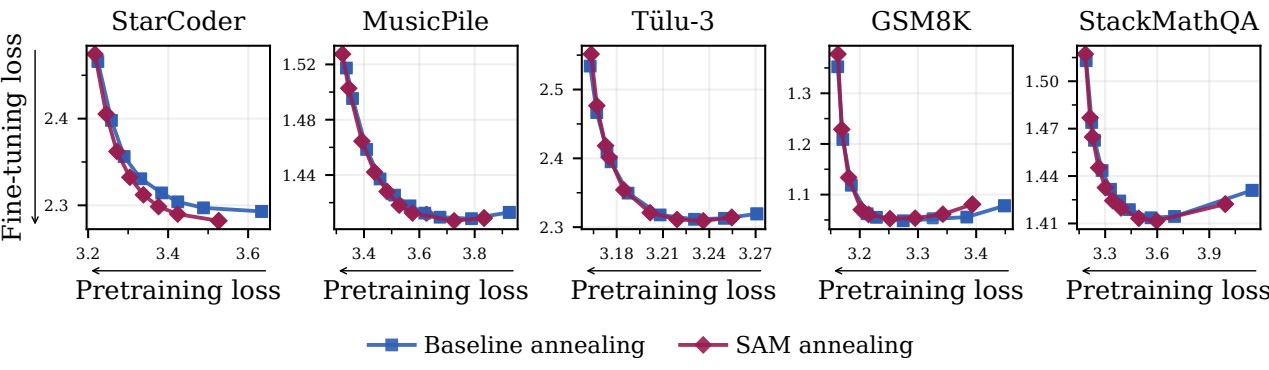

*Figure 55.* **Annealing with SAM vs. baseline annealing (WSD) learning-forgetting frontier across datasets for OLMo-150M pretrained on 120B tokens.**

E.4.2. GAUSSIAN PERTURBATION SENSITIVITY

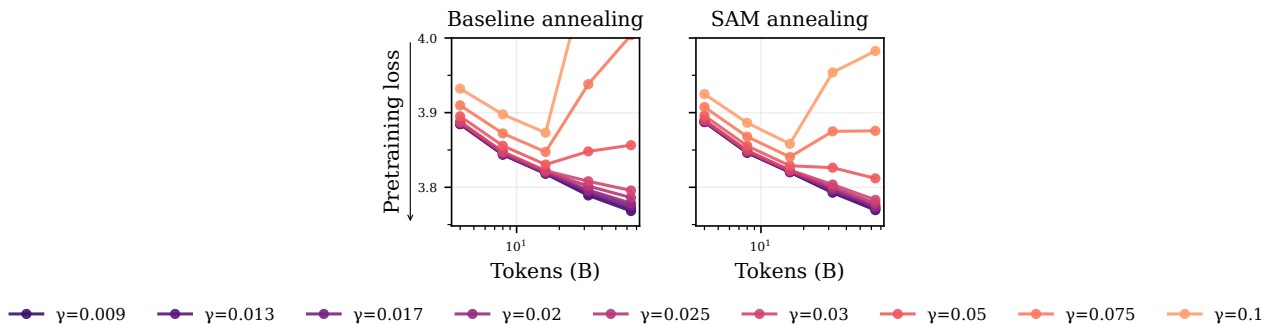

*Figure 56.* **Annealing with SAM vs. baseline annealing (WSD) pretraining loss vs. pretraining tokens at different perturbation magnitudes $\gamma$ for OLMo-20M.**

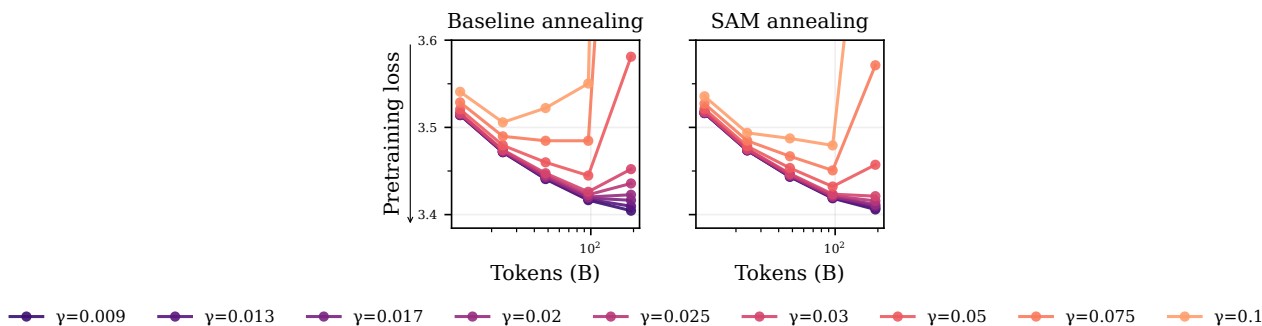

*Figure 57.* **Annealing with SAM vs. baseline annealing (WSD) pretraining loss vs. pretraining tokens at different perturbation magnitudes $\gamma$ for OLMo-60M.**

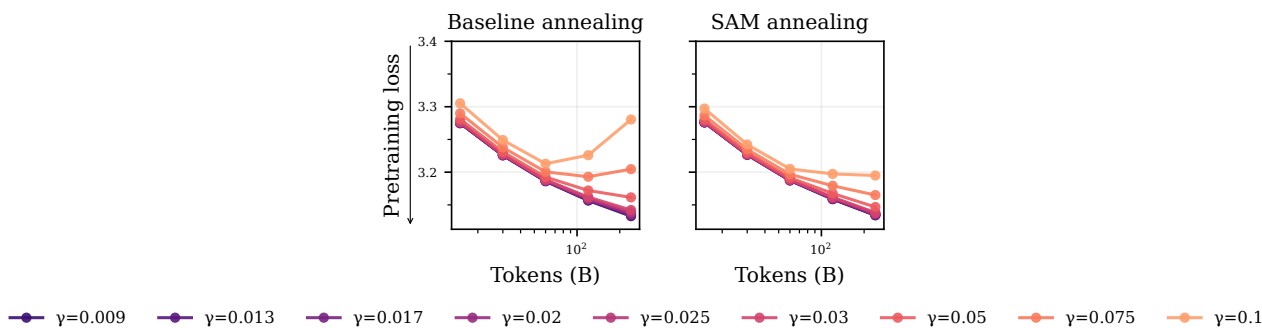

*Figure 58.* **Annealing with SAM vs. baseline annealing (WSD) pretraining loss vs. pretraining tokens at different perturbation magnitudes $\gamma$ for OLMo-150M.**

### E.4.3. POST-TRAINING QUANTIZATION PERFORMANCE

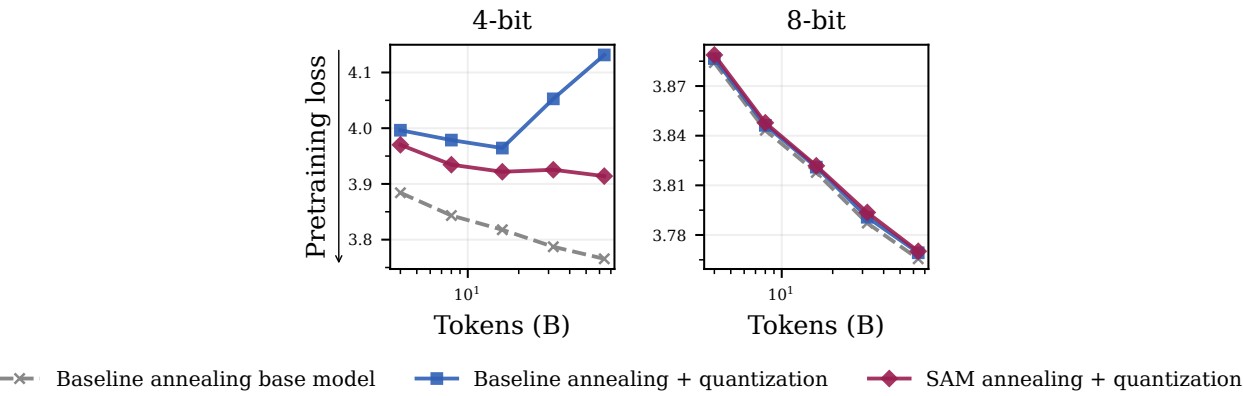

*Figure 59.* **Annealing with SAM vs. baseline annealing (WSD) under 4-bit and 8-bit post-training quantization for OLMo-20M.**

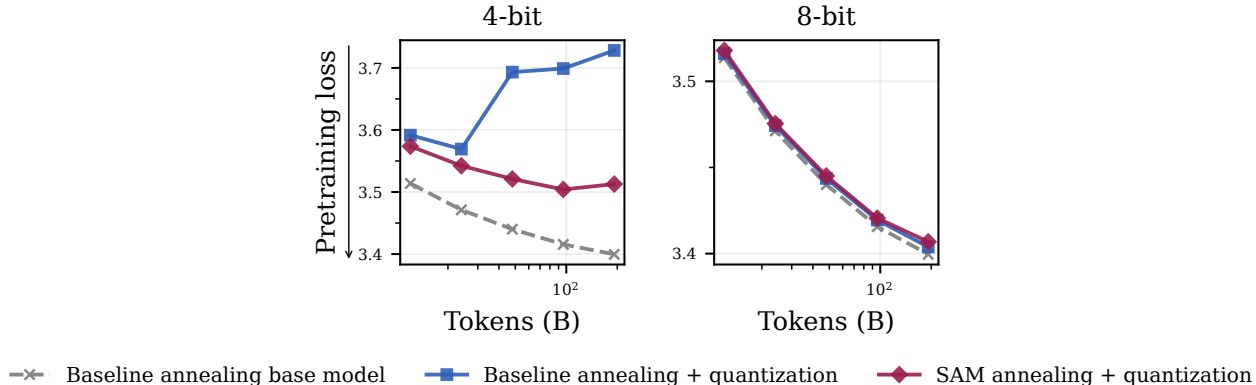

*Figure 60.* **Annealing with SAM vs. baseline annealing (WSD) under 4-bit and 8-bit post-training quantization for OLMo-60M.**

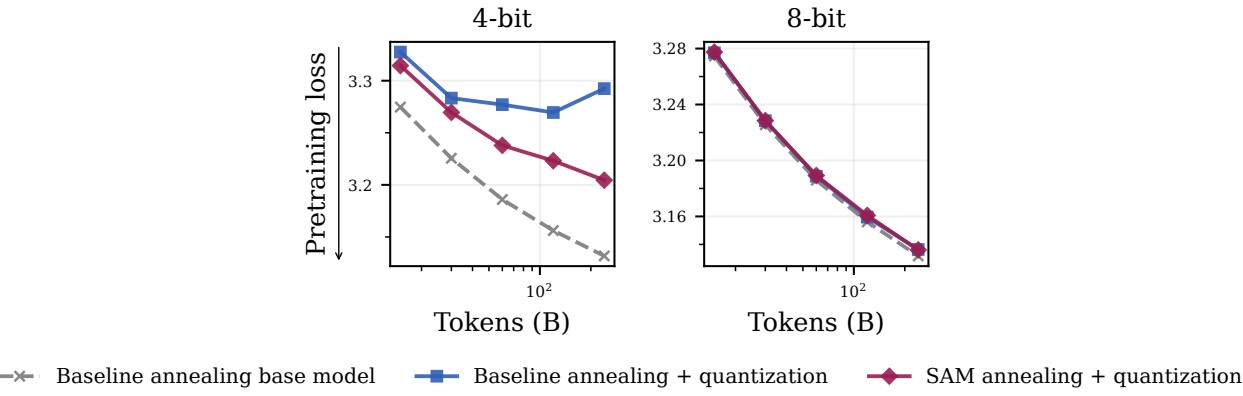

*Figure 61.* **Annealing with SAM vs. baseline annealing (WSD) under 4-bit and 8-bit post-training quantization for OLMo-150M.**

# F. Analysis of the Hessian

We have thus far established that optimization methodologies that explicitly and implicitly minimize sharpness can yield a superior learning-forgetting tradeoff. In this section, we return to our original intuition and ask, *to what extent does the minimization of sharpness explain this improvement?*

In Section 2, we made two assumptions to motivate SAM and the increase of the peak learning rate.

1. **Loss admits a second-order Taylor approximation under fine-tuning perturbations;** this implies the Hessian governs the extent of performance degradation under such perturbations.

2. **The recipes reduce fine-tuning-direction curvature;** whether explicitly or implicitly, these methods minimize curvature in the fine-tuning direction.

Neither assumption is guaranteed. Fine-tuning may move far enough that higher-order terms dominate. Likewise, small-batch SAM can reduce Hessian trace without reducing fine-tuning-direction curvature (Wen et al., 2023b), and high learning rates can constrain spectral norm without minimizing fine-tuning-directional sharpness (Cohen et al., 2021; Damian et al., 2023).

We thus test both assumptions, using OLMo-60M checkpoints fine-tuned on StarCoder.

### F.1. Does local sensitivity explain fine-tuning degradation?

We first explore the degree to which the second-order Taylor expansion,

$$\mathcal{L}_{\text{PT}}(\theta_{\text{FT}}) \approx \mathcal{L}_{\text{PT}}(\theta_{\text{PT}}) + \tfrac{1}{2}\Delta_{FT}^{\top} H \Delta_{FT}, \tag{11}$$

holds for fine-tuning perturbations $\Delta_{FT}$. For each StarCoder fine-tuning run, we compare observed post-fine-tuning pretraining loss with the quadratic prediction across token budgets, fine-tuning learning rates, and both optimizers. Here $\Delta_{\text{FT}}$ is the actual fine-tuning update associated with each particular run.

**The quadratic approximation (roughly) upper bounds loss.** Across learning rates and token budgets, the quadratic prediction usually overestimates post-fine-tuning loss (Figure 62). Reducing this term should therefore reduce an empirical upper bound on forgetting.

**Pretraining learning rates tighten the approximation; fine-tuning learning rates loosen it.** When sweeping the peak pretraining learning rate for AdamW at a fixed 192B-token budget (Figure 63), the quadratic approximation continues to upper bound the observed loss, with the tightest bounds occurring at higher peak learning rates. In contrast, when sweeping fine-tuning learning rates, the approximation remains tight for small learning rates but becomes increasingly loose as the fine-tuning learning rate grows.

**The quadratic approximation worsens at larger token budgets and under AdamW.** The quality of the quadratic approximation degrades as token budgets increase: it tightly upper bounds the observed loss at small budgets but becomes a loose upper bound at larger budgets (Figure 62). Additionally, models trained with SAM are more accurately captured by this approximation than those trained with AdamW.

### F.2. How well is fine-tuning-directional sharpness minimized?

We next test whether the recipes reduce directional sharpness, $\Delta_{\text{FT}}^{\top} H \Delta_{\text{FT}}/\|\Delta_{\text{FT}}\|^2$, along the fine-tuning perturbation. We measure it at a canonical fine-tuning learning rate of $4 \times 10^{-4}$ and plot it against pretraining tokens, both for the SAM-vs-AdamW comparison and for the peak-learning-rate sweep (Figure 64).

**Models trained with SAM show reduced directional sharpness.** Consistent with prior literature, we find that as models are trained on more tokens, the sharpness of the loss in the fine-tuning direction increases progressively (Springer et al., 2025; Cohen et al., 2021). Models trained with SAM exhibit a slower increase in directional sharpness than their AdamW counterparts, while also maintaining consistently lower directional sharpness (Figure 64.a). Note that $\Delta_{FT}$ is endogenous to the optimizer being compared, so directional sharpness here is a property of the (base, fine-tuning) pair rather than the base model alone.

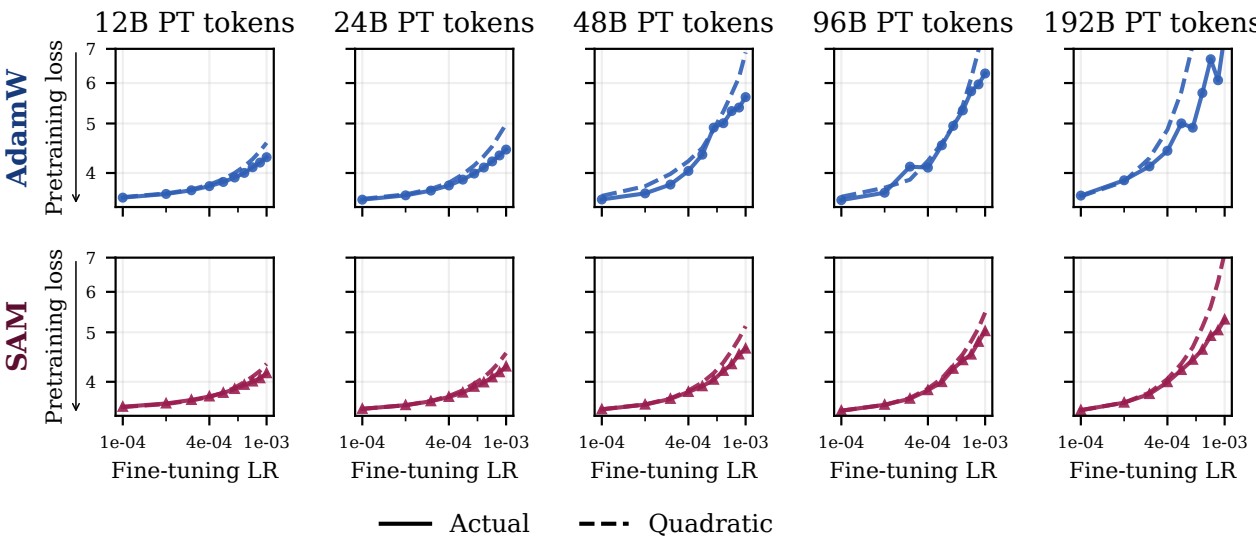

*Figure 62.* **Quadratic approximation vs. observed loss for the token sweep.** We compare 60M AdamW and SAM checkpoints fine-tuned on StarCoder after pretraining on 12B, 24B, 48B, 96B, and 192B tokens. Columns correspond to token budget, the top row shows AdamW, and the bottom row shows SAM. Solid lines show the observed pretraining loss after fine-tuning as we sweep the fine-tuning learning rate, and dashed lines show the quadratic approximation (Equation 11).

**Larger peak pretraining learning rates reduce directional sharpness.** We find a similar relationship in the peak-learning-rate sweep at 192B tokens: larger peak learning rates reduce directional sharpness (Figure 64.b). The reduction is monotonic across the rates we sweep.

The degradation of the loss after fine-tuning is determined by the product of the directional sharpness and the (squared) distance traveled during fine-tuning. This implies that the reduced sensitivity induced by SAM, in fact, arises from the flattening of the base model along the direction of fine-tuning, rather than from any reduction in the fine-tuning step size.

> **Summary:** SAM and higher peak learning rates reduce directional sharpness along the fine-tuning direction, which empirically upper-bounds and explains their gains in post-fine-tuning loss.

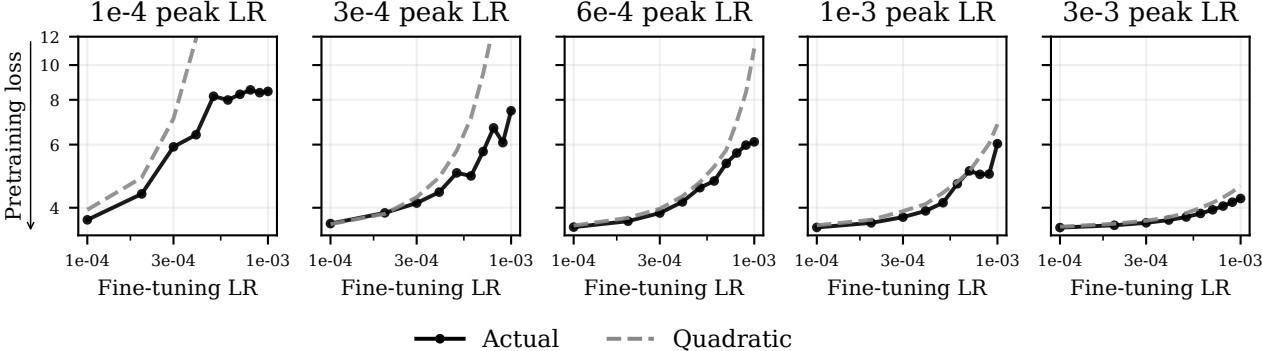

*Figure 63.* **Quadratic approximation vs. observed loss across fine-tuning learning rates.** We fix 60M AdamW checkpoints at 192B pretraining tokens and fine-tune on StarCoder. Each panel corresponds to a different peak pretraining learning rate. Solid lines show the observed pretraining loss after fine-tuning as we sweep fine-tuning learning rate, and dashed lines show the quadratic approximation.

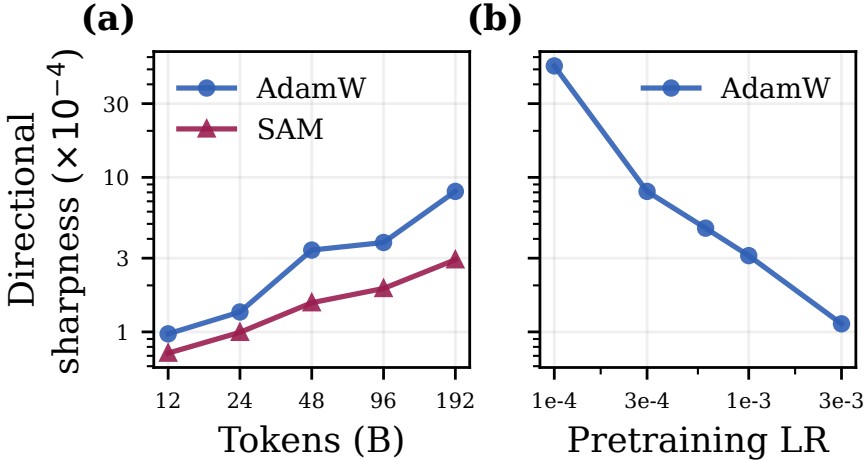

*Figure 64.* **SAM and large peak learning rates both lower fine-tuning directional sharpness.** For 60M StarCoder checkpoints fine-tuned with learning rate $4 \times 10^{-4}$, **(a)** normalized directional sharpness vs pretraining tokens for AdamW and SAM at their canonical peak learning rates, and **(b)** normalized directional sharpness vs pretraining peak learning rate for AdamW at 192B tokens.

