# OpenReview forum: "Sharpness-Aware Pretraining Mitigates Catastrophic Forgetting"
_ICML.cc/2026/Conference — ICML 2026 regular_

### Official Review · Reviewer_RMR9 · 2026-03-03

**Soundness:** 3
**Presentation:** 4
**Significance:** 4
**Originality:** 3
**Overall Recommendation:** 5
**Confidence:** 5

**Summary:**

This paper explores the question of how we might continue to pre-train LLMs to very high tokens-per-parameter (TPP) ratios, while not incurring a catastrophic loss of plasticity, or a catastrophic forgetting, or other accuracy degradations whether catastrophic or just unfortunate.  The proposed solution is what in classical ML has been called "regularization": accepting a quality degradation on the training distribution in order to generalize better to unseen data, but here the "generalization" is not accuracy on a held-out set or different domain, but accuracy after the base model weights are fine-tuned, quantized, or perturbed.  The regularization in this paper is achieved via either sharpness-aware minimization (SAM), or by trying higher learning rates (higher peaks or higher sustained periods).  The robustness to weight changes are viewed through the lens of loss landscape sharpness. Across different model scales, (very high) token budgets, and datasets, we find sharpness-aware pre-training results in a better learning/forgetting frontier in SFT, and higher accuracy after quantization or weight perturbation.  The idea of training with a higher LR for most of training, but then using an explicit SAM optimizer (with higher compute overhead) during the decay portion only, is proposed and evaluated.

**Compliance With Llm Reviewing Policy:**

Affirmed.

**Final Justification:**

Taking into account both the paper and the authors' rebuttal, I increased my score for Soundness (the authors proposed improvements will help) and thereby increased my overall recommendation to "Accept."  Good paper.

**Key Questions For Authors:**

Why do Figure 6 left and right have different x-axes, and why does the Cosine result on the right not match one of the Cosine results on the left?  Which Cosine are you comparing to on the right?  I.e., what minimum LR did you use with Cosine and WSD?

Weight averaging is often proposed as an alternative to decaying the LR.  E.g., DeepSeekV3 averaged checkpoints on CPU to track progress throughout training.  Does model averaging somehow increase sharpness too?  Or is that an even better approach than large learning rates and WSD, enabling high-quality base models without sharpness?

WSD was originally pitched as a way to check loss at various intermediate training milestones, decaying the LR in a separate branch in order to check intermediate model progress with a decayed LR.  Wen et al argued (with experimental results) not to "waste" these decay branches, but to resume training after the decay.  This makes me wonder, could we somehow just warm up the LR on the original pre-training distribution BEFORE doing FT or downstream eval (implicit sharpness decrease)?  I see you do warmup on the FT dataset but maybe that’s not good enough?  Should we do a small SAM phase (explicit sharpness decrease) on the PT distribution?

Are there lessons from Dohare, "Loss of plasticity in deep continual learning", 2024, that are worth addressing in the paper?  You discuss loss of plasticity but not prior methods to deal with it.  A more fullsome discussion of related work could be very helpful to readers here.

**Limitations:**

No.  As I noted above:

It is rare to use monolithic distributions for PT and SFT in modern LLM practice.
- Mid-training and data curriculums are used extensively in PT.
- Replay is used extensively to prevent forgetting in SFT.

These aren't fatal problems, but you should have an explicit limitations section that discusses them in detail.

There are indubitably other limitations that the authors would do well to address in a specific limitations section ("we only tried one model architecture, or dataset type, or vocabulary, we didn't try linear decay, we didn't try chaning the weight decay or batch size, etc., etc.").  Could even go in the appendix if you reference it in the main paper.

**Strengths And Weaknesses:**

Strengths:

Overall this is a very exciting paper, and I feel could be very impactful.  The idea of avoiding sharp regions of the loss landscape makes a lot of sense, and seems like a natural follow-up to prior work demonstrating catastrophic overtraining.  The results as presented are fairly convincing that sharpness-awareness, or making a base model more robust prior to SFT or quantization, is a good idea.

The writing, figures, takeaways, organization, etc., are all stellar.

Weaknesses:

Mainly I am not convinced there's anything special about WSD versus Cosine, which is a major claim of this paper.  The reasons are as follows: the paper tunes the LR for the Cosine schedule's validation loss.  This same LR is used for the WSD schedule.  The paper clearly establishes that higher LRs can generalize better.  If we use the same peak LR for WSD, it will have a higher average LR overall.  So is the advantage of the WSD schedule the shape/functional form, or is it the higher LR?  I don't think the paper answers this question.

"We tune the learning rate for our baseline (AdamW+cosine) model checkpoints to minimize the pre-training validation loss, then reuse these learning rates for all other settings. This
choice implicitly favors the baseline and ensures our results are not driven by unfair tuning."

This would be true *unless higher LRs are systematically better*, which is precisely what you show.  In this case, the WSD schedule will have an advantage.  Imagine you compared two schedules: Cosine and Cosine-2x, where Cosine-2x is exactly the same except it uses twice the LR of the Cosine schedule.  You tune LR for "Cosine" on the PT distribution, but you find Cosine-2x is better across all FT and quantization tests.  Would you conclude Cosine-2x is a better schedule, after all, you implicitly favored Cosine?  No, they are the same schedule!  This is how I feel about WSD as currently presented.  (I believe WSD *could* have some inherent advantage because maybe the key is to have a higher LR later in training... but the fact 5% to 20% decay phases perform similarly makes the think that even this is unlikely).

I tried to figure this out: maybe you compare WSD to a Cosine schedule that was tuned for SFT?  Maybe in Figure 6?  But I don’t really understand why Figure 6 left and right have different x-axes, and why the Cosine curve on the right doesn’t match one of the Cosine curves on the left.  Which Cosine are you comparing to on the right?

Other comments/suggestions:

The impact of the work is surely diminished because as far as I know, it is rare to use monolithic distributions for PT and SFT in modern LLM practice.
- Mid-training and data curriculums are used extensively in PT.
- Replay is used extensively to prevent forgetting in SFT.

These aren't fatal problems, but you should have an explicit limitations section that discusses them in detail.

The Additional Results in the appendix are rather too extensive.  I mean, 128 figures seems like a bit much to just process sequentially.  Have you considered maybe summarizing the findings from these plots into either a single summary figure or at least a summary table?  If you made a summary table, with hyperlinks to the figures, we could drill down where we have interest.  But better would be to collect a summary statistic from each figure, and then help us understand how these summary statistics evolve as certain things vary.  I.e., make a summary plot.


The idea of doing worse on the PT distribution in order to generalize better is precisely the definition of regularization in classical ML.  I would have liked to see this discussed more in this context.  Here we have a different kind of regularization that focuses on making the models amenable to perturbation.  But what about standard generalization, as in, what about just negative log-likelihood loss on the different validation sets?  I mean, did we need to do the FT on the validation sets to discover the value of SAM and larger LRs?

Minor:

Figure 6: "For the Cosine-based models (left), we see that a higher learning rate with suboptimal pretrain-loss achieves better forgetting." --- this is a little confusing I think because Figure 6 shows PT loss AFTER FT, whereas I think you mean it has a lower PT loss BEFORE FT (after PT), which I’m not sure "we see" in the figure?  Anyways I found this confusing.


Typos:
- “using Sharpness-Aware Minimization … to explicitly penalize sharpness during pre-training effectively [??] and improves downstream performance”
- Figure 7 caption: What’s the dashed line?
- Figure 7 caption: “peretrain loss”, “gaussian”

---

> ### Author Rebuttal · Authors · 2026-03-31
>
> Thank you for recognizing our contributions. We've added new large-scale experiment results using data-curriculum and interventions to mitigate forgetting during SFT, addressing limitations and broadening the paper's scope. We also clarified our analysis of WSD, which you rightly noted was flawed.
>
> >Mainly I am .... this question.
>
> We hypothesize that WSD performs better due to a shorter effective annealing period, which keeps the learning rate (LR) higher for longer. Experiments with OLMo-60M show WSD annealing percentages (5, 10, 20, 50, 100)  yield better Pareto performance than Cosine [Figure 1: Anneal-Percent](https://anonymous.4open.science/r/icml26-rebuttal-1C45/Anneal_Percent/pareto.png). However, we don’t test whether this is the *only* reason for improvement (i.e., if some other aspect of the shape matters), and will revise our draft to clearly state when we find WSD to be better.
>
> >"We tune .... unfair tuning."
>
> >This would .... this is unlikely).
>
> We believe you are absolutely right. Our intent was to compare default, commonly used training choices (e.g., cosine schedules, as used in prior work like OLMo) against alternatives such as WSD, rather than exhaustively re-tuning each schedule. We found that for a given setting of Max and Min LR, WSD utilizes it more efficiently [Figure 2: Peak-LR](https://anonymous.4open.science/r/icml26-rebuttal-1C45/Peak_LR/pareto_starcoder.png). However, we agree that to compare two schedules, we should tune them for SFT instead and will include this in the final revision.
>
> >I tried to figure .... the right?
>
> Figure 6 has no discrepancy; both plots use the same cosine curve, though the x-axis scale differs, which we will fix. We used the same learning rate (3e-4) for both WSD and cosine in these plots.
>
> >The impact .... in detail.
>
> We present further results on an OLMo2-1B model trained with the standard OLMo recipe and data curriculums. A checkpoint pretrained on 4T tokens (~4000 TPP) was midtrained using SAM/AdamW for 50B tokens. We evaluate these models on downstream accuracy tasks (e.g., GSM8k, Arc-Challenge, MMLU, Hellaswag, etc.) and report the average forgetting after PTQ and SFT. Table 1 shows that using SAM, even for a short period, improves forgetting, demonstrating the scalability of our interventions to real-world scenarios.
>
> | Optimizer | PTQ | SFT (Meta-Math) | SFT (StackMathQA) | SFT (Tulu) |
> |-----------|-----|-----------------|-------------------|------------|
> | AdamW     | 8.6% | 3.5%           | 3.0%              | 3.8%       |
> | SAM       | 5.4% **(36.9% ↑)** | 1.6% **(55.9% ↑)** | 1.9% **(34.7% ↑)** | 2.4% **(38.2% ↑)** |
>
> Table 1
>
> We also share Pareto frontier for the 1B model, analogous to Fig. 4 [Figure 3: Pareto-1B-SFT-MetaMath](https://anonymous.4open.science/r/icml26-rebuttal-1C45/OLMo2-1B/meta-math_pareto_1b.png). While Data Replay wasn't tested during SFT, we experimented with Elastic Weight Consolidation (EWC) [1], a continual learning technique using the Fisher Information matrix to finetune in directions least sensitive to the pretraining distribution. We found EWC reduces forgetting, and SAM further improves it! [Figure 4: SAM+EWC](https://anonymous.4open.science/r/icml26-rebuttal-1C45/EWC/pareto_60m_starcoder_192b.png)
>
> >The idea .... larger LRs?
>
> Our primary goal is not to measure generalization in isolation but to clarify the relationship between our work and the long line of work on regularization for generalization. Emphasizing post-finetuning performance is essential, as it captures effects like forgetting and robustness not fully reflected by pretraining validation loss alone.
>
> >Figure 6 .... found this confusing.
>
> We agree the current phrasing is confusing and will fix this in the revision.
>
> >Weight averaging .... without sharpness?
>
> That's a great question, and would be good for future work.
>
> >WSD was originally .... PT distribution?
>
> That's also a great question! This is roughly our intuition behind SAWD (only applying SAM during annealing time) — by applying SAM only at the end of training, even if the model had, before activating SAM, found a sensitive minima, SAM might be able to recover from this. It would be very interesting to explore how best to "re-warm" an already sharp language model.
>
> >Lessons Dohare, .... helpful to readers here.
>
> Dohare et al. propose explicit structural reinitialization to preserve diversity, while our work uses landscape sharpness as a geometric proxy for such internal failures. [2] show that SAM improves feature quality by suppressing overly dominant features and countering the simplicity bias of SGD, leading to more diverse representations. Building on these perspectives, we argue our approach mitigates forgetting not only by finding flat regions, but also by implicitly maintaining high-rank, balanced feature representations as highlighted in Dohare.
>
> **References:**
>
> [1] Overcoming catastrophic forgetting in neural networks
>
> [2] SAM Enhances Feature Quality via Balanced Learning

---

> > ### Author Rebuttal · Reviewer_RMR9 · 2026-04-01
> >
> > The authors have agreed that "to compare two schedules, we should tune them for SFT instead and [they] will include this in the final revision."  This was my main concern and I suppose it *will* be resolved.  That is, maybe there's nothing special about WSD, assuming it's tuned equivalently to Cosine.  Beyond that, I appreciate the studies with the data curriculums, and sure, why not, elastic weight consolidation.
> >
> > By the way, when I said, "what about just negative log-likelihood loss on the different validation sets?", I meant:
> > - compare whether SAM or higher LRs allow us to generalize better to new domains, e.g., if we compute log-likelihood on the DOWNSTREAM data sets, do we do better?  This would obviate the need for FT... but yes I agree it wouldn't capture forgetting, and so I kind of agree your setup is necessary.

---

### Official Review · Reviewer_17fX · 2026-03-05

**Soundness:** 3
**Presentation:** 4
**Significance:** 3
**Originality:** 2
**Overall Recommendation:** 4
**Confidence:** 4

**Summary:**

This paper proposes SAM for the last stage pretraining, which improves over the standard methods for the learning-forgetting setup. This works particularly well when paired with WSD scheduler.

**Compliance With Llm Reviewing Policy:**

Affirmed.

**Final Justification:**

The authors acknowledged the need to better position their work within the literature, plus provided new results showing the accuracy metric instead of loss.

**Key Questions For Authors:**

The empirical results are solid, and the contribution is interesting. My main concern with the paper is that it does not adequately position itself within the literature. This is important to help the readers (and reviewers) better evaluate the originality of the contributions. Also, I would like to see the accuracy in addition to the loss. If the authors can properly contextualize their work and provide the new result, I will reconsider the evaluation.

**Limitations:**

yes

**Strengths And Weaknesses:**

Pros

- The paper is well written and presented.
- The idea is simple and it is relevant for the pretraining and finetuning setup.
- The evaluation setup seems solid. I have checked Appendix B.1 and B.2, which cover in detail the hyperparameter sweep.
- Figure 2 shows that SAM helps more for finetuning at high TPP, which is interesting.

Cons

- The paper would benefit from a more in-depth discussion of the implicit bias of large LR training in the related work. For example, discussing these papers [1, 2, 3, 4, 5].
- The authors should improve the presentation of the results by discussing them with a broader literature. Currently, different sections read as a technical report.
- The idea of the paper seems to greatly overlap with [6], altho the authors focus mainly on the learning forgetting tradeoff in LM models. Can the authors discuss their work with [6]?
- The authors should also at least discuss (or even better compare) with existing SAM (flatness) methods applied in a continual learning setup, which is highly related. I am not familiar with that field, but as a quick search, one example is [7].
- Why does SAM help more for finetuning at high TPP? Does the author have any intuition on this phenomenon?
- I would also like to see accuracy instead of only loss, since lower loss does not always correspond to better downstream tasks.
- Typo: L270 use \` and \' for quoting lagging behind.
- The impact section goes over the 8 page limit.

[1] https://arxiv.org/abs/2210.05337

[2] https://arxiv.org/abs/1907.04595

[3] https://arxiv.org/abs/2302.07011

[4] https://openreview.net/forum?id=RU76KTF1Da

[5] https://openreview.net/forum?id=ZXr3Xx7Z1O

[6] https://arxiv.org/abs/2410.10373

[7] https://arxiv.org/abs/2404.00986

---

> ### Author Rebuttal · Authors · 2026-03-31
>
> We thank the reviewer for acknowledging our scientific contributions. We have added a new, exciting, large-scale experiment with accuracy evaluations and incorporated discussion on relevant literature to better position our work, which we briefly discuss here.
>
> >The paper would .... these papers [1, 2, 3, 4, 5].
>
> This is really good feedback! We acknowledge that a more in-depth discussion is required and will incorporate them in the final revision.
>
> >The idea of the paper .... their work with [6]?
>
> Our work is complementary to [6] but focuses on a different question. While [6] shows that applying SAM only at the end of training is sufficient to find flat minima, our work studies how different mechanisms—specifically SAM and the inductive bias induced by large learning rates—affect the learning–forgetting tradeoff in language models.
>
> Notably, [6] does not consider fine-tuning or the learning–forgetting dynamics we analyze. We do agree with the observation that applying SAM at the end of training can be effective; [6] establishes this in terms of flatness, and we further show that such flatness correlates with a favorable learning–forgetting tradeoff. We will cite [6] appropriately and clarify its connection in our revision, and explain how this perspective helps account for the success of applying SAM at the end of training in our setting.
>
> >The authors should .... search, one example is [7].
>
> We agree this is a relevant line of work and will discuss it in the revision. [7] studies continual learning in the classical sequential task setting (e.g., learning a sequence of tasks with smaller-scale models such as ResNet architectures), where the goal is to mitigate catastrophic forgetting across tasks by promoting flat minima. In contrast, our work focuses on LLM pretraining and fine-tuning, where the pretraining distribution is expected to support downstream adaptation rather than a sequence of disjoint tasks. Both approaches share the idea that flatness improves robustness, but they target different regimes and problem settings.
>
> Our goal is to improve the learning–forgetting tradeoff from the pretraining stage itself, making models inherently more robust for downstream use. We view this as complementary to continual learning methods like C-Flat. In particular, as per Reviewer PaNu's guidance, we ran additional experiments using another popular continual learning technique, Elastic Weight Consolidation (EWC) [1]. We observed that SAM still provides gains, with SAM+EWC outperforming AdamW+EWC, suggesting that SAM provides benefits complementary to those in classical CL methods [Figure 1: SAM+EWC](https://anonymous.4open.science/r/icml26-rebuttal-1C45/EWC/pareto_60m_starcoder_192b.png). We will include a discussion of [7] and clarify these differences in the revision.
>
> >Why does SAM help .... intuition on this phenomenon?
>
> Our intuition is that SAM helps models at all TPP find a less-sensitive minima. Language models that have been trained with a very high TPP (with AdamW) have been shown to be especially sensitive to parameter perturbations such as finetuning [2] and quantization [3], and thus the effect of SAM on learning and forgetting is more pronounced in this regime.
>
> >I would also like to see accuracy .... downstream tasks.
>
> We report additional results on an OLMo2-1B [4] model trained using the standard OLMo recipe. We began with a checkpoint pretrained on 4T tokens (approximately 4000 TPP) and then midtrained it using SAM/AdamW for 50B tokens. We evaluate these models on downstream accuracy tasks (e.g., GSM8k, Arc-Challenge, MMLU, Hellaswag, etc.) and report the average forgetting, which is the percentage decrease in accuracy from the midtrained model after PTQ and SFT.
>
> We tuned SFT for multiple learning rates, similar to the small-scale experiments, and recorded the average forgetting at a thresholded finetuning loss. As shown in Table 1, we observe that using SAM, even for this small period of training, reduces forgetting by over 34%. This highlights its practical relevance and demonstrates that our interventions scale to real-world scenarios.
>
> | Optimizer | PTQ | SFT (Meta-Math) | SFT (StackMathQA) | SFT (Tulu) |
> |-----------|-----|-----------------|-------------------|------------|
> | AdamW     | 8.6% | 3.5%           | 3.0%              | 3.8%       |
> | SAM       | 5.4% **(36.9% ↑)** | 1.6% **(55.9% ↑)** | 1.9% **(34.7% ↑)** | 2.4% **(38.2% ↑)** |
>
> Table 1: Percentage of forgetting (lower is better) after PTQ and SFT on OLMo2-1B, mid-trained with SAM versus AdamW.
>
> We also share a Pareto frontier analogous to Fig. 4 for the 1B model [Figure 2: Pareto-1B-SFT-MetaMath](https://anonymous.4open.science/r/icml26-rebuttal-1C45/OLMo2-1B/meta-math_pareto_1b.png).
>
> **References:**
>
> [1] [1612.00796] Overcoming catastrophic forgetting in neural networks
>
> [2] [2503.19206] Overtrained Language Models Are Harder to Fine-Tune
>
> [3] [2411.04330] Scaling Laws for Precision
>
> [4] [2501.00656] 2 OLMo 2 Furious

---

> > ### Author Rebuttal · Reviewer_17fX · 2026-04-02
> >
> > Thanks for the interesting rebuttal, especially the EWC part.
> > - What hyperparameter of EWC is used across the setups?
> > - Regarding the downstream accuracy task, only the relative improvement is reported, but what is the absolute accuracy? I need to understand if the baseline is significant

---

> > > ### Author Response · Authors · 2026-04-03
> > >
> > > Thank you for considering our rebuttal. We find that the gains of SAM over AdamW hold for each value of the hyperparameter $\lambda$ [Figure 3: EWC Tuning](https://anonymous.4open.science/r/icml26-rebuttal-1C45/EWC/pareto_60m_starcoder_192b_tuning.png) of the EWC (and consequently the best [Figure 4: SAM+EWC](https://anonymous.4open.science/r/icml26-rebuttal-1C45/EWC/pareto_60m_starcoder_192b_final.png)).
> > >
> > > For the 1B models with midtraining, our baseline roughly matches the Olmo results (refer to Table 9 [1]).
> > > We believe the baselines are in fact significant and the gains are real and show up even at a larger scale. We note that we only used SAM during mid-training; we expect the gains to be further amplified when performing SAM during the entire pretraining.
> > >
> > > We hope these sufficiently addressed your two main concerns.
> > >
> > > More details below:
> > >
> > > >What hyperparameter of EWC is used across the setups?
> > >
> > > The EWC loss is given as,
> > > $L(\theta) = L_{new}(\theta) + \lambda \sum_i F_i (\theta_i - \theta_i^*)^2$
> > >
> > > - $L_{new}(\theta) $: Loss for the current task (e.g., cross-entropy)
> > > - $\theta_i$: Current parameter
> > > - $\theta^*_i$: Parameter after training on the previous task
> > > - $F_i$: Importance of parameter *i*, estimated using the Fisher Information Matrix
> > > - $\lambda$: Strength of regularization (hyperparameter)
> > >
> > > We plot the pareto using the following hyperparameters:
> > > - **Learning rates:**  [1e-4, 1.5e-4, 2e-4, 2.5e-4, 3e-4, 3e-5e-4, 4e-4, 5e-4, 6e-4, 7e-4, 8e-4, 1e-3]
> > > - **Loss Regularization ($\lambda$):** 1e+3, 4e+3, 1e+4*, 4e+4, 1e+5
> > > - **Samples to compute Fisher Information Matrix (F):** 100 batches, batch size 64 and sequence length 1024. ~6.5M pretrain tokens.
> > >
> > > >Regarding the downstream accuracy task, only the relative improvement is reported, but what is the absolute accuracy? I need to understand if the baseline is significant
> > >
> > > We report the entire table of downstream evals for 1B models on top of which we had reported percentage improvements in Table 1.
> > >
> > > | | Average | ARC_C | BoolQ | HellaSwag | MMLU | Winogrande | DROP | NQ_Open | AGIEval | GSM8K | MMLU_Pro | TriviaQA |
> > > |--------|--------|-------|-------|-----------|------|------------|------|---------|---------|-------|-----------|----------|
> > > | **AdamW (Midtrained)** | 45.95 | 47.40 | 73.30 | 69.40 | 44.30 | 67.40 | 33.40 | 20.90 | 34.80 | 44.00 | 15.40 | 55.10 |
> > > | **SAM (Midtrained)**   | 45.85 | 47.10 | 74.90 | 69.20 | 44.40 | 67.30 | 32.90 | 20.40 | 35.70 | 42.50 | 15.50 | 54.40 |
> > >
> > > Table 2: Downstream eval scores for 1B models midtrained on 50B tokens using SAM and AdamW
> > >
> > > **References:**
> > >
> > > [1] [2501.00656] 2 OLMo 2 Furious

---

### Official Review · Reviewer_PaNu · 2026-03-07

**Soundness:** 2
**Presentation:** 4
**Significance:** 3
**Originality:** 3
**Overall Recommendation:** 4
**Confidence:** 4

**Summary:**

This paper studies the impact of optimization and hyperparameter choice in pretraining on catastrophic forgetting in downstream fine-tuning after pretraining. They first provide a theoretical justification that sharpness is a means to measure an approximation for forgetting, which motivates their design for an optimizer termed SAM, which searches for a minima which retains a low loss given a small perturbation. They rigorously test their idea on models of 20-150M parameters and 12B-192B tokens. They additionally explore the effects of learning rate, scheduling, and decay on sharpness and the ability to mitigate forgetting during finetuning.

**Compliance With Llm Reviewing Policy:**

Affirmed.

**Final Justification:**

The rebuttal addressed my main concerns. I wished for larger-scale pretraining results; however, those are outside the time constraints. The authors answered the remainder of my questions. I recommend they be accepted.

**Key Questions For Authors:**

1. How does SAM hold when scaled up to much larger models (e.g., 350M and 750M)?
2. Does SAM improve and aid finetuning method which explicity attempt to mitigate forgetting?
3. SAM is only used during the D part of WSD, and you further show in Figure 6 that a smaller D phase reduces the curvature of the final model. Do the benefits of SAM become negligible as you decrease the WSD decay rate down to 5% or lower, as it appears to minimally affect final training loss?

**Limitations:**

Yes

**Strengths And Weaknesses:**

Strengths
- The justification of the importance of finding a flat minimum in training to mitigate catastrophic forgetting is justified and clear.
- The overall flow and presentation of the paper are well structured and clear.
- While in general, the curvature (e.g., Hessian and Fisher Information Matrix) is well known to heavily affect catastrophic forgetting (e.g., EWC), to my knowledge, the analysis of reducing the curvature of the final model during pretraining seems novel.
- The robustness of the hyperparameter searches for finetuning is particularly convincing, and adds to the strength of the method. Additionally, the hyperparameter section is sound for the Adam v. SAM comparison.

Weaknesses
- The experiments are limited to very small models, all under 150M. Analysis of larger models (e.g., 350M, 770M) would greatly strengthen the paper.
- The combination of SAM with finetuning methods, which explicitly only optimize directions with low curvature (e.g., EWC) during finetuning, has not been done. Several works have shown that large-scale pretraining already moves to relatively flat loss basins in pretraining, so it is not apparent the impact that additional "flatness" that SAM adds when used in combination with methods that finetune only flat directions.

---

> ### Author Rebuttal · Authors · 2026-03-31
>
> We thank the reviewer for acknowledging our scientific contributions. We have added a new, exciting, large-scale experiment and also explained how our findings further benefit continual learning techniques.
>
> >The experiments are limited to very small models, all under 150M. Analysis of larger models (e.g., 350M, 770M) would greatly strengthen the paper.
>
> >How does SAM hold when scaled up to much larger models (e.g., 350M and 750M)?
>
> We report additional results on an OLMo2-1B [1] model trained using the standard OLMo recipe. We began with a checkpoint pretrained on 4T tokens (approximately 4000 TPP) and then midtrained it using SAM/AdamW for 50B tokens. We evaluate these models on downstream accuracy tasks (e.g., GSM8k, Arc-Challenge, MMLU, Hellaswag, etc.) and report the average forgetting, which is the percentage decrease in accuracy from the midtrained model after PTQ and SFT.
>
> We tuned SFT for multiple learning rates, similar to the small-scale experiments, and recorded the average forgetting at a thresholded finetuning loss. As shown in Table 1, we observe that using SAM, even for this small period of training, reduces forgetting by over 34%. This highlights its practical relevance and demonstrates that our interventions scale to real-world scenarios.
>
> | Optimizer | PTQ | SFT (Meta-Math) | SFT (StackMathQA) | SFT (Tulu) |
> |-----------|-----|-----------------|-------------------|------------|
> | AdamW     | 8.6% | 3.5%           | 3.0%              | 3.8%       |
> | SAM       | 5.4% **(36.9% ↑)** | 1.6% **(55.9% ↑)** | 1.9% **(34.7% ↑)** | 2.4% **(38.2% ↑)** |
>
> Table 1: Percentage of forgetting (lower is better) after PTQ and SFT on OLMo2-1B, mid-trained with SAM versus AdamW.
>
>
> We also share a Pareto frontier analogous to Fig. 4 for the 1B model [Figure 1: Pareto-1B-SFT-MetaMath](https://anonymous.4open.science/r/icml26-rebuttal-1C45/OLMo2-1B/meta-math_pareto_1b.png).
>
> >The combination of SAM with finetuning methods, which explicitly only optimize directions with low curvature (e.g., EWC) during finetuning, has not been done. Several works have shown that large-scale pretraining already moves to relatively flat loss basins in pretraining, so it is not apparent the impact of additional "flatness" that SAM adds when used in combination with methods that finetune only flat directions.
>
> >Does SAM improve and aid fine tuning methods which explicitly attempt to mitigate forgetting?
>
> This is a great question and we have run additional experiments on 60M OLMo models to directly address this. We find that SAM + EWC does better than AdamW +EWC. We find that while EWC does reduce forgetting, the benefits of SAM persist! [Figure 2: SAM+EWC](https://anonymous.4open.science/r/icml26-rebuttal-1C45/EWC/pareto_60m_starcoder_192b.png). We are very excited about this result and will incorporate it into the main paper.
>
> >SAM is only used during the D part of WSD, and you further show in Figure 6 that a smaller D phase reduces the curvature of the final model. Do the benefits of SAM become negligible as you decrease the WSD decay rate down to 5% or lower, as it appears to minimally affect final training loss?
>
> We identify in this work that spending more time in the lower learning rate "annealing" portion is responsible for incurring forgetting. However, it has been noted (or prevailing intuition) is that annealing is important to get good pretrain performance in the first place [2]. So there's an interplay between the two: too less annealing, the pretrain loss is not low enough - both SAWD and AdamW would perform similarly but poorly. SAWD allows us to do annealing and learn the pretraining distribution better while not incurring the increased forgetting that AdamW does.
>
> **References**
>
> [1] [2501.00656] 2 OLMo 2 Furious
>
> [2] Understanding Warmup-Stable-Decay Learning Rates: A River Valley Loss Landscape Perspective

---

> > ### Author Rebuttal · Reviewer_PaNu · 2026-04-02
> >
> > Thank you for the response. I appreciate the mid-training and EWC results; they more or less answer my questions. It would be nice to see from scratch pre-training results for larger models; however, I understand this is unreasonable in a rebuttal timeline.
> >
> > I maintain my score.

---

> > > ### Author Response · Authors · 2026-04-03
> > >
> > > Thank you for the thoughtful feedback! Even applied to just a small fraction of training steps (mid-training), SAM yields strong gains, consistent with the same phenomenon observed at smaller scales. We expect these to further amplify with from-scratch pre-training.
> > >
> > > Given that your primary concerns have been addressed, we would appreciate reconsideration of the score and are happy to clarify any remaining questions.

---

### Official Review · Reviewer_VMBb · 2026-03-08

**Soundness:** 3
**Presentation:** 3
**Significance:** 3
**Originality:** 3
**Overall Recommendation:** 5
**Confidence:** 4

**Summary:**

The authors hypothesize that lower loss sharpness is related to better fine-tuning performance and better quantization robustness of the models after pretraining. Motivated by this, they propose using Sharpness-Aware Minimization (SAM), Warmup-Stable-Decay (WSD) and their combination Sharpness-Aware Warmup-Stable-Decay (SAWD) to improve these measures. They show that indeed these measures are improved substantially for highly overtrained small (20M, 60M, 150M) OLMo2-like models.

**Compliance With Llm Reviewing Policy:**

Affirmed.

**Final Justification:**

The authors added OLMo2-1B model experiments, and as promised in the review I am raising my score to 5. As I said in the review, I am not sure about the applicability of the proposed methods to real pretraining but the findings are important scientifically.

**Key Questions For Authors:**

1. Could the authors clarify what "Compute Matched" figures in the Appendix show, and what is the interpretation? Is this explained anywhere?

2. Related to the piece "Modern training often operates near a critical edge of stability where learning rate and Hessian eigenvalues jointly govern dynamics (Cohen et al., 2021)", do the authors know of any literature more relevant to their setting, and what precisely happens in their setting?

**Limitations:**

Yes

**Strengths And Weaknesses:**

**Strengths**:
- The research question is very reasonable. It is conjectured based on previous related work such as (Springer et al., 2025) that loss sharpness is connected to fine-tuning performance, so it is natural to try sharpness-aware minimization.

- After carefully going through all experiment results, I believe the conclusions. The effect is moderate (and likely too moderate to be practically useful), but it is there. In particular, authors indeed connect sharpness with fine-tuning performance and showcase the correctness of their intuitions. This is likely useless (in this form) for improving pretraining, but the results are very valuable scientifically. In other words, the intervention does not work too well, and has overhead that makes it unjustified, but the world needs to know that it works at all.

**Weaknesses**:

TPP = tokens per parameter
- The first claim of the paper is that SAM achieves better learning-forgetting frontiers compared to AdamW when using the cosine learning rate schedule. This is true of the 60M model trained at the absurdly large 3200 TPP (Fig. 1). As a point of minor importance, the picture is pretty mixed for the smaller 20M model. More importantly, the picture is mixed for less absurd 200 TPP (Fig. 22) and for the larger model: for the 150M model at 100 TPP, the outperformance of SAM is questionable (Fig. 32). I would suspect that the authors' claims are true but require extremely large token horizons to manifest themselves.

- The same is true of the comparison of learning rate schedules. Looking only at the largest 150M model at 100 TPP (Fig. 36), I do not clearly see outperformance of the proposed learning rate schedules (WSD or SAWD).

- The same is true of the gains for robustness: SAM improves sensitivity to perturbations and quantization primarily in very high TPP regimes, as authors point out (Fig. 7 and end of Sec. 4.4).

- SAM has very large overhead. Applying it during the whole pretraining run is likely not justified because this doubles the required pretraining compute, whereas benefits are only pronounced enough at extremely high token horizons. Authors admit this, and advocate using SAM only for the decay phase in WSD. However, looking at the largest model, I don't see much benefit of SAWD compared to WSD at any token horizons (Fig. 36 to 39). In short, with compute efficiency taken into account, the proposed methods are unlikely to improve pretraining.

- The largest model (150M) is still very small, and I suspect essentially no conclusions are importantly true at 20-100 TPP. I would really love to see an experiment like Figure 4 but with larger models (and perhaps lower TPP). This may cause me to raise the score, if sufficiently convincing and unless other reviewers identify weaknesses I have not identified. I will ignore how much importance other reviewers assign to the ones I already have identified. (I am willing to defend this paper based on the scientific contribution rather than the usefulness of the method.)

---

> ### Author Rebuttal · Authors · 2026-03-31
>
> We thank you for acknowledging the value of our scientific contribution. We have added a new, exciting large-scale experiment and also explain how our experiments address a regime of practical interest.
>
> We report additional results on an OLMo2-1B [1] model trained using the standard OLMo recipe. We started with a checkpoint pretrained on 4T tokens (~4000 TPP) and then midtrained it using SAM/AdamW for 50B tokens. We evaluate these models on downstream accuracy tasks (e.g., GSM8k, Arc-Challenge, MMLU, Hellaswag, etc.) and report the average forgetting, which is the percentage decrease in accuracy from the midtrained model after PTQ and SFT.
>
> We tuned SFT for multiple LRs, similar to the small-scale experiments, and recorded the average forgetting at a thresholded finetuning loss. In Table 1, we observe that using SAM, even for this small period of training (analogous to SAWD), reduces forgetting by over 34%. This highlights the practical relevance and demonstrates that our interventions scale to real-world scenarios.
>
> | Optimizer | PTQ | SFT (Meta-Math) | SFT (StackMathQA) | SFT (Tulu) |
> |-----------|-----|-----------------|-------------------|------------|
> | AdamW     | 8.6% | 3.5%           | 3.0%              | 3.8%       |
> | SAM       | 5.4% **(36.9% ↑)** | 1.6% **(55.9% ↑)** | 1.9% **(34.7% ↑)** | 2.4% **(38.2% ↑)** |
>
> Table 1: Percentage of forgetting (lower is better) after PTQ and SFT on OLMo2-1B, mid-trained with SAM versus AdamW.
>
> >The first claim of the paper is that SAM .... horizons to manifest themselves.
>
> >The same is true of the comparison of learning rate .... rate schedules (WSD or SAWD).
>
> >The same is true of the gains for robustness: .... out (Fig. 7 and end of Sec. 4.4).
>
> We agree that the gains from SAM and our proposed schedules become more pronounced at larger token-per-parameter (TPP) budgets. However, as noted above, the TPP budgets we consider are standard in modern LLM training. For example: OLMo2-1B [1] (4T tokens => ~ 4000 TPP), LLaMA-3 8B [2] (15T tokens => ~ 2000 TPP), Gemma-3 1B [3] (2T tokens => ~ 2000 TPP), and Qwen models 0.7B-32B [4] (~ 36T tokens). These trends indicate that the regimes where our improvements are seen are still of practical interest. We also note that at smaller TPP, SAM remains competitive, does not degrade performance, and offers improved robustness.
>
> >The largest model (150M) is still very small, .... willing to defend this paper based on the scientific contribution rather than the usefulness of the method.)
>
> As noted before, we have shared experiments at the 1B scale using the standard LLM training recipe used by OLMo, which show the usefulness of our approach in practical settings. To directly address this concern, we also share a Pareto frontier analogous to Fig. 4 for the 1B model. [Figure 1: Pareto-1B-SFT-MetaMath](https://anonymous.4open.science/r/icml26-rebuttal-1C45/OLMo2-1B/meta-math_pareto_1b.png).
>
> >SAM has a very large .... are unlikely to improve pretraining.
>
> This is a great question. One of the main goals of our work is to show that both SAM and WSD (via reduced annealing) can yield improvements in certain regimes. SAWD shows substantial gains over WSD in downstream settings such as quantization, which is a widely used inference-time modification. Regarding Figures 36–39, the 150M results are reported at a much lower TPP (800) compared to the 20M and 60M results (3200 TPP), which may limit the observed benefits at that scale. We hypothesize that SAM's advantages become more pronounced at higher TPP budgets and do not diminish with scale, a claim backed by our compute efficient mid-training results on the 1B model.
>
> >Could the authors clarify what "Compute Matched" figures in the Appendix show, and what is the interpretation? Is this explained anywhere?
>
> We apologize for not including a detailed description of the setup and will add it in the final revision. Since SAM typically requires 2x the compute per token, to match compute for these figures, we pre-train the SAM models on half the tokens compared to AdamW. Even under this constraint, SAM outperforms AdamW at high TPP (a common scenario in modern LLM training) (see Figures 60–64 for 60M and 74–79 for 150M). We also show that SAWD (where the expensive SAM step is applied only at the very end of training) achieves similar benefits at a substantially lower compute cost.
>
> >Related to the piece "Modern training .... happens in their setting?
>
> Cohen et al. show that a large learning rate has an implicit bias that leads to flat solutions. [5] explains that SAM can, in fact, minimize the maximum eigenvalue or the trace of the eigenvalues of the Hessian, also leading to flat solutions, which is analogous to the findings of Cohen et al.
>
> **References**
>
> [1] [2501.00656] 2 OLMo 2 Furious
>
> [2] [2407.21783] The Llama 3 Herd of Models
>
> [3] [2503.19786] Gemma 3 Technical Report
>
> [4] [2505.09388] Qwen3 Technical Report
>
> [5] [2211.05729] How Does Sharpness-Aware Minimization Minimize Sharpness?

---

> > ### Author Rebuttal · Reviewer_VMBb · 2026-04-03
> >
> > Thank you for interesting responses! The OLMo2 experiments are indeed valuable. For the record, such large TPP counts are standard usually for small versions of models, not flagship versions I think.

---

### Decision · Program_Chairs · 2026-04-30

**Decision:**

Accept (regular)

**Comment:**

This paper proposes using SAM during the decay phase of the WSD scheduler during pre-training to reduce catastrophic forgetting during finetuning. All the reviewers agree that this is a good paper to accept (even though there are some concerns about how practically useful the results are -reviewer VMBb). All the reviewers agree that this paper is technically sound.

Reviewer 17fX highlighted several relevant papers that this paper fails to cite or discuss. I would also like to add [1], which has already shown that SAM reduces forgetting in pre-trained models. In general, the literature review seems to be the main weakness of this paper, and I strongly encourage the authors to include all relevant papers in their discussion in the final version.

[1] https://www.jmlr.org/papers/v24/22-0496.html